# EHMT2 methyltransferase governs cell identity in the lung and is required for KRAS [G12D] tumor development and propagation

Ariel Pribluda[1]*[†], Anneleen Daemen[2][‡], Anthony Nelson Lima[1], Xi Wang[1], Marc Hafner[2], Chungkee Poon[3], Zora Modrusan[4], Anand Kumar Katakam[5], Oded Foreman[5], Jefferey Eastham[5], Jefferey Hung[5], Benjamin Haley[4], Julia T Garcia[6], Erica L Jackson[7][§], Melissa R Junttila[1]*[‡]

[1]Department of Translational Oncology, Genentech, Inc, South San Francisco, United States; [2]Department of Bioinformatics & Computational Biology, Genentech, Inc, South San Francisco, United States; [3]Department of Immunology, Genentech, Inc, South San Francisco, United States; [4]Department of Molecular Biology, Genentech, Inc, South San Francisco, United States; [5]Department of Pathology, Genentech, Inc, South San Francisco, United States; [6]Department of Genetics, Stanford University, Stanford, United States; [7]Department of Discovery Oncology, Genentech, Inc, South San Francisco, United States

*For correspondence:
ariel@surrozen.com (AP);
melissa.junttila@oricpharma.com
(MRJ)

Present address: [†]Surrozen, South San Francisco, United States; [‡]ORIC Pharmaceuticals, South San Francisco, United States; [§]Scorpion Therapeutics, South San Francisco, United States

**Abstract** Lung development, integrity and repair rely on precise Wnt signaling, which is corrupted in diverse diseases, including cancer. Here, we discover that EHMT2 methyltransferase regulates Wnt signaling in the lung by controlling the transcriptional activity of chromatin-bound β-catenin, through a non-histone substrate in mouse lung. Inhibition of EHMT2 induces transcriptional, morphologic, and molecular changes consistent with alveolar type 2 (AT2) lineage commitment. Mechanistically, EHMT2 activity functions to support regenerative properties of Kras[G12D] tumors and normal AT2 cells—the predominant cell of origin of this cancer. Consequently, EHMT2 inhibition prevents Kras[G12D] lung adenocarcinoma (LUAD) tumor formation and propagation and disrupts normal AT2 cell differentiation. Consistent with these findings, low gene EHMT2 expression in human LUAD correlates with enhanced AT2 gene expression and improved prognosis. These data reveal EHMT2 as a critical regulator of Wnt signaling, implicating Ehmt2 as a potential target in lung cancer and other AT2-mediated lung pathologies.

## Editor's evaluation

This is generally a well-designed and carefully conducted study that is likely to be of interest to many both inside and outside of the field of lung tumorigenesis and normal lung development and tissue homeostasis.

## Introduction

Lung cancer is the leading cause of cancer mortality in men and women, surpassing combined deaths from colon, prostate and breast cancer (https://www.cancer.org/, 2019). Approximately 40% of non-small cell carcinoma (NSCLC) is of the adenocarcinoma subtype. Genomic alterations of KRAS and

TP53 mutations are the most frequent events within this tumor subtype (*Cancer Genome Atlas Research, 2014*). Considerable evidence indicates that alveolar type 2 (AT2) cells are the predominant cell of origin of lung adenocarcinoma (LUAD) (*Desai et al., 2014*; *Mainardi et al., 2014*; *Sutherland et al., 2014*; *Xu et al., 2012*).

Sustained tumor growth is maintained, in some instances, by a subset of plastic, stem-like cells, referred to as tumor-propagating cells (TPCs) (*Batlle and Clevers, 2017*; *Beck and Blanpain, 2013*). This rare cell subset is less sensitive to therapeutic intervention, and functionally contributes to tumor regrowth following treatment responses (*Beck and Blanpain, 2013*). TPCs are hypothesized to evolve extensively from the tumor-initiating event (*Reya et al., 2001*; *Shackleton et al., 2009*), yet the fidelity to its cell of origin remains undefined. TPCs in LUAD have been previously characterized and were shown to be required for Kras$^{G12D}$ tumor self-renewal (*Zheng et al., 2013*). Moreover, a TPC gene signature correlates with poor prognosis in non-small cell LUAD patients (*Zheng et al., 2013*). While these data implicate a critical function for TPCs in driving tumor progression, it is unknown how these rare cells maintain their stem-like properties.

Wnt signaling is of critical importance in the lung, controlling tissue development, homeostasis and repair processes following lung damage. Wnt signaling is critical in the distal airway to maintain AT2 cell fate and function in the alveolar compartment. A subset of AT2 cells serve as adult tissue stem cells, replenishing themselves, as well as alveolar type 1 (AT1) cells, which are both required to maintain proper alveolar function. Deletion of β-catenin in AT2 cells leads to AT1 cell fate differentiation, demonstrating that β-catenin-mediated Wnt signaling is required for AT2 cell identity (*Frank et al., 2016*; *Nabhan et al., 2018*). Along the same lines, genetic models have also shown the forced expression of a stabilized form of β-catenin engenders aberrant cell fate in mouse lung (*Pacheco-Pinedo et al., 2011*). Together these data indicate that precise Wnt pathway activity is important for alveolar cell function, as well as, cell fate decisions. Furthermore, disruption of Wnt signaling equilibrium can have major consequences in some disease pathologies, including lung cancer (*Nabhan et al., 2018*; *Tammela et al., 2017*; *Zacharias et al., 2018*). What remains unclear is how Wnt signals are intrinsically regulated in cells to maintain or acquire facultative stem-like properties in both homeostatic and disease contexts. Understanding the mechanisms that underlie cell fate decisions may enable therapeutic approaches to enhance or prevent these processes for patient benefit.

In many biological contexts, cellular self-renewal and lineage fate commitment is controlled by epigenetic mechanisms, such as chromatin remodeling (*Easwaran et al., 2014*; *Hemberger et al., 2009*; *Widschwendter et al., 2007*). Recent work identified expression of the EHMT2 lysine methyltransferase (G9a) as a poor prognostic factor in LUAD (*Chen et al., 2010*; *Huang et al., 2017*). Interestingly, EHMT2 activity has been implicated in the regulation of cell identity during development primarily through its epigenetic histone methyltransferase activity (*Chen et al., 2012*; *Epsztejn-Litman et al., 2008*). While EHMT2 methyltransferase activity and its regulation of chromatin is well-described; the non-histone targets of epigenetic regulators are emerging as critical signaling mediators. Here, we investigated how EHMT2 methyltransferase functions to regulate cell fate in distinct cellular contexts relevant for lung tumor development, propagation and AT2 biology. We discover a mechanism of cell intrinsic Wnt signaling control that is governed by EHMT2 activity, establishing EHMT2 as a crucial arbitrator of cell fate gene expression in the lung. By discovering this alternative cell-intrinsic mechanism of governing Wnt signaling in cells, we are able to manipulate cell fate decisions to predictably limit the functionality of these cells to disable tumor propagation, development and AT2 trans-differentiation.

## Results

### EHMT2 activity is required for *Kras$^{G12D}$;Trp53* (KP) tumorsphere self-renewal

Given the association of EHMT2 expression and poor prognosis in LUAD (*Chen et al., 2010*; *Huang et al., 2017*), we sought to examine the expression and function of EHMT2 in primary murine *Kras$^{G12D}$;Trp53$^{-/-}$* (KP) tumors. Previous work established that KP tumor self-renewal was dependent on a TPC subset. Therefore, by utilizing the previously characterized surface markers CD24, ITGB4, and NOTCH (*Zheng et al., 2013*), we sorted the TPC population and evaluated EHMT2 protein expression. Using two distinct detection methods we observed a consistent and robust increase in EHMT2 protein

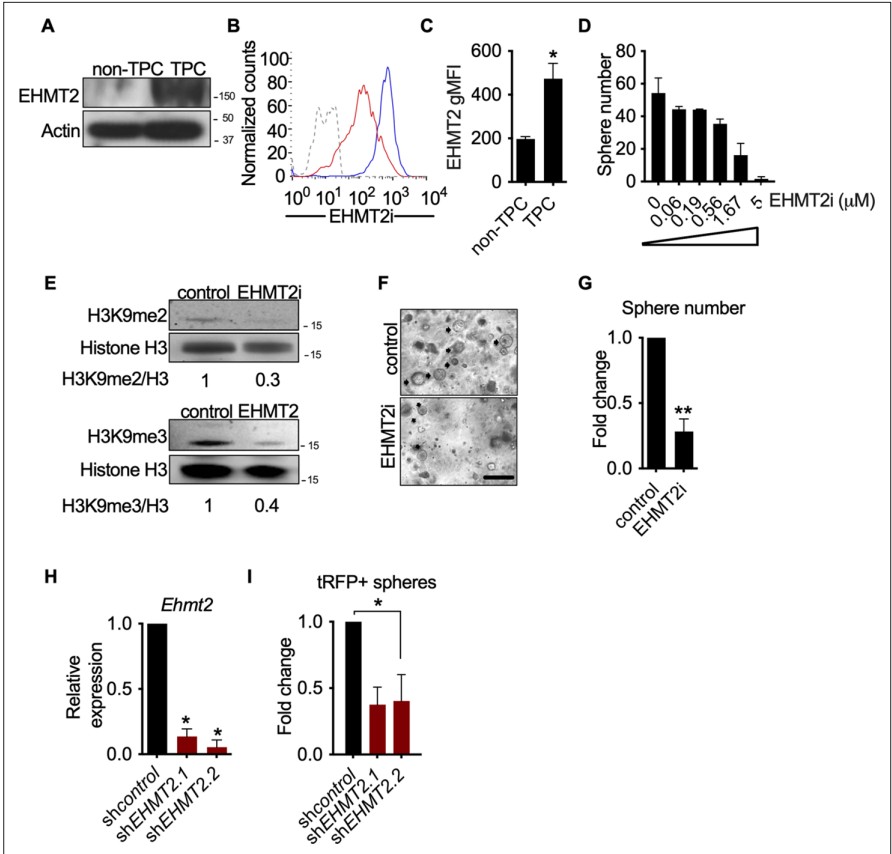

**Figure 1.** EHMT2 activity is required for *Kras*[G12D];*Trp53* (KP) tumorsphere self-renewal. (**A**) Western blot analysis of EHMT2 in tumor-propagating cell (TPC) and non-TPC. Actin was used as loading control. (**B**) Flow cytometry analysis of EHMT2 in TPC and non-TPC (blue, TPC; red, non-TPC; gray, Isotype). (**C**) Quantification of EHMT2 geometric fluorescence intensity (gMFI) in (**B**) (n=2, mean ± SEM; two-tailed t-test, *p=0.05). (**D**) Tumorsphere formation of KP-derived primary cells seeded with increasing doses of EHMT2 inhibitor (n=2, mean ± SEM, One-way ANOVA with multiple testing, *p<0.005). (**E**) Western blot analysis showing reductions in H3K9me2/3 following EHMT2 inhibitor treatment (EHMT2i, EHMT2 inhibition) histone H3 was used as loading control. Ratio of H3K9me to H3 is depicted at the bottom of the Western blot. (**F**) Representative image of primary tumorspheres following secondary passaging in the absence of either vehicle control or EHMT2 inhibitor (EHMT2i, EHMT2 inhibition. Scale bar 100 μm). (**G**) Quantitation of tumorsphere growth after secondary passaging (n=5; mean ± SEM; two-tailed paired t-test, **p<0.005). (**H**) Relative qRT-PCR of *Ehmt2* transcripts from primary tumorspheres, expressing either short hairpin RNA (shRNA) control (shControl), or shRNAs against *Ehmt2* (sh*Ehmt2.* 1, sh*Ehmt2.* 2), (n=2; mean ± SEM; two-tailed paired t-test, *p<0.05). (**I**) Quantification of turbo RFP (tRFP)-positive tumorspheres following secondary passage of primary tumorspheres expressing control or *Ehmt2* shRNAs. (sh*Ehmt2*.1, n=2; mean ± SD, sh*Ehmt2*.2, n=3; mean ± SD; two-tailed paired t-test, *p<0.05, shEHMT2.1 p=0.09).

The online version of this article includes the following source data and figure supplement(s) for figure 1:

**Source data 1.** Western blot for *Figure 1A* showing G9a (EHMT2) in non-TPC vs TPC.

**Source data 2.** Western blot (right side) for *Figure 1A* showing ACTIN in non-TPC vs TPC.

**Figure supplement 1.** Schematic overview of tumorsphere assay.

expression (*Figure 1A–C*). Next, we evaluated the requirement of EHMT2 in self-renewal function of TPCs using ex vivo KP-derived organotypic cultures (i.e. tumorspheres), an established surrogate for measuring the in vivo regenerative capability of TPCs (*Zheng et al., 2013*). Pharmacological inhibition of EHMT2, using UNC0642 (*Liu et al., 2013*), resulted in a dose-dependent decrease of primary ex vivo KP-derived tumorsphere formation (*Figure 1D*). Pharmacological inhibition of EHMT2 in established tumorspheres led to a marked reduction in both histone H3 lysine 9 di- and tri-methylation marks (H3K9me2/3), consistent with potent EHMT2 inhibition (*Collins and Cheng, 2010*; *Epsztejn-Litman et al., 2008*; *Shinkai and Tachibana, 2011*; *Figure 1—figure supplement 1*). Both UNC0642

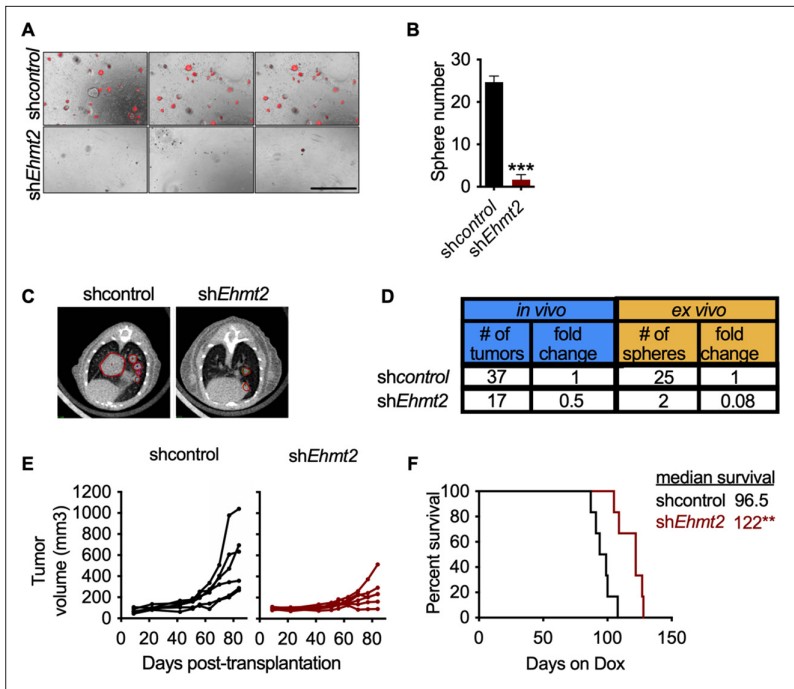

**Figure 2.** EHMT2 is required for in vivo tumor self-renewal. (**A**) Ex vivo analysis of tumorsphere formation from primary orthotopic transplanted cells, expressing either short hairpin RNA (shRNA) control (sh*control*) or shRNAs against *Ehmt2* (sh*Ehmt2*) (n=3 ± SD) Scale bar 500µM. (**B**) Quantification of tumorsphere formation in panel (**A**), two tailed t-test, \*\*\*p<0.0005 (**C**) Representative µ-CT images of shcontrol- or sh*Ehmt2*-expressing tumors (n=6) red circles depicting tumors. (**D**) Table comparing efficiencies of secondary passage in vivo and ex vivo from orthotopically-transplanted primary KP cells, expressing sh*control* (n=6) or sh*Ehmt2* (n=6). (**E**) Overall tumor volume in secondary recipient mice orthotopically transplanted with KP cells from primary recipients, expressing either shRNAs targeting control (sh*control*) or *Ehmt2* (sh*Ehmt2.1*) (n=6) tumor volume at end of study by treatment, Mann-Whitney test, \*p<0.05. (**F**) Graph indicates survival of mice depicted in (**E**) (n=6 per group. Geha-Breslow-Wilcoxon test, \*\*p<0.005).

The online version of this article includes the following figure supplement(s) for figure 2:

**Figure supplement 1.** Analysis of in vivo serial orthotopic transplantation of primary KP cells harboring *Ehmt2* targeting hairpins.

**Figure supplement 2.** Analysis of terminal tumors from in vivo serial orthotopic transplantation of primary KP cells harboring *Ehmt2* targeting hairpins.

treatment or short hairpin RNA (shRNA)-mediated depletion of *Ehmt2* similarly impaired secondary sphere formation (*Figure 1F–I*), albeit only shEHMT2.2 reached statistical significance, establishing a requirement for EHMT2 activity in TPC self-renewal.

## EHMT2 is required for in vivo tumor growth

Previous work has demonstrated that serial re-growth of KP tumors following orthotopic transplantation requires a sustained functional TPC population (*Zheng et al., 2013*). To evaluate the functional necessity of EHMT2 activity on in vivo tumor formation, we evaluated tumor formation following serial transplantation of orthotopically transplanted KP-derived primary cells harboring doxycycline (Dox)-inducible shRNAs. First, hairpin expression was induced in vivo for 13 days by doxycycline administration to mice with established primary lung tumors. Thereafter, shRNA-expressing cells were sorted from primary tumors and assessed for secondary tumorsphere formation ex vivo and tumor formation in vivo (*Figure 2—figure supplement 1A-C*). Sorted sh*Ehmt2*-expressing cells from primary recipients showed a significant decrease in both ex vivo tumorspheres and in vivo secondary tumor formation, establishing a role for EHMT2 in maintaining TPC stemness (*Figure 2A–D*). Continuous monitoring of secondary transplants in vivo revealed a substantial growth impairment in sh*Ehmt2*-expressing tumors, which translated to a significant increase in overall survival (*Figure 2E–F*). Mice harboring

sh*Ehmt2*-expressing tumors eventually succumb to tumor outgrowth; however, analysis of terminal tumors revealed re-expression of *Ehmt2* transcript to a level equivalent to that of control tumors (*Figure 2—figure supplement 2A,B*). This data demonstrates that *Ehmt2* expression is required for TPC-tumor growth. Taken together, these data indicate that EHMT2 activity in TPCs functions to maintain the self-renewal capacity of KP tumors.

## EHMT2 preserves TPC function by preventing AT2 differentiation

To elucidate the mechanistic basis of EHMT2 in maintaining tumor self-renewal, we characterized the phenotypic impact of pharmacological inhibition of EHMT2 in tumorspheres. EHMT2 inhibition resulted in significant reductions in BrdU-labeled cells (fivefold) and expression of cleaved caspase-3 (>10 fold) (*Figure 3—figure supplement 1A-D*). Reduced proliferation and cell death were previously associated with cell differentiation (*Domen and Weissman, 1999*; *Ruijtenberg and van den Heuvel, 2016*), therefore, we further explored cell fate changes as a possible treatment outcome. Since LUAD arises predominantly from the distal alveolar compartment (*Mainardi et al., 2014*; *Sutherland et al., 2014*; *Xu et al., 2012*), we quantified established gene signatures pertaining to distal alveolar cell types from RNA sequencing data derived from EHMT2 i-treated tumorspheres (*Treutlein et al., 2014*). The AT2 gene signature was significantly increased following EHMT2 inhibition, in contrast to other cell lineages (*Figure 3A*). Protein expression of surfactant protein C (SPC), a canonical AT2 marker was also significantly upregulated in EHMT2 -inhibited tumorspheres, likely due to differentiated progenitors (*Figure 3—figure supplement 2*). Concomitant with an increase in SPC+, we observed an increase in CD74, an additional cell surface marker characterizing AT2 cells (*Lee et al., 2013*), showing an increase of the double-positive population (1.8-fold) (*Figure 3B and C*). Moreover, we observed a significant increase in the transcript levels of multiple surfactants in both EHMT2-inhibited and EHMT2-depleted (sh*Ehmt2*) tumorspheres (*Figure 3—figure supplement 3A.B*), indicating increased/enhanced AT2-like cell fate features when EHMT2 activity is impaired. Importantly, this AT2-like conversion was confirmed in *Ehmt2*-depleted tumor cells from secondary passage in vivo (*Figure 3—figure supplement 4*). Furthermore, evidence supporting cell fate transition was observed when transmission electron microscopy (TEM) of EHMT2-inhibited tumorspheres revealed a significant increase in lamellar bodies (*Balis and Conen, 1964*), which are distinct specialized structures responsible for the storage and release of surfactants and serve as a morphometric readout for AT2 cells (*Figure 3D and E*). Together these results demonstrate that EHMT2 inhibition triggers an enhanced/reinforced AT2-like cell state in KP-derived tumorspheres.

As EHMT2 activity is critical for the self-renewal of tumorspheres and its expression is selectively enriched in TPCs, we interrogated whether the observed cell fate changes occur within the TPC fraction. EHMT2 inhibition in primary tumorspheres caused a reduction in TPCs, (*Figure 3—figure supplement 5*), consistent with their reduced stemness. Notably, EHMT2-inhibited TPCs displayed a statistically significant increase (fourfold) in the AT2 surface markers, SPC and CD74, compared to matched controls (*Figure 3F and G*). Consistently, *Sftpc* and *Cd74* transcripts increased exclusively in TPCs (*Figure 3H*). Together, these data indicate that EHMT2 inhibition leads to an induced AT2-like cell state within the TPC subset, thereby reducing their regenerative capacity by a mechanism similar in features to differentiation. To assess whether the relationship between EHMT2 activity and cell state extends to human tumors, we used clinical adenocarcinoma specimens and assessed the association between *EHMT2* transcript and cell lineage gene signatures (*Treutlein et al., 2014*) in a panel of 546 LUADs (*Cancer Genome Atlas Research, 2014*). Consistent with our murine data, *EHMT2* gene expression negatively correlates with the AT2 cell gene signature (*Figure 3—figure supplement 6*), supporting the concept that EHMT2 activity impairs differentiation. Taken together, the data indicate that EHMT2 activity represses an alveolar differentiation program in murine LUAD as a means to preserve stem-like properties that enable tumor self-renewal.

## Wnt activation impairs TPC self-renewal and induces AT2 cell lineage marker expression

Previous work indicates that EHMT2 can suppress promiscuous transcription by regulating chromatin structure through the positioning of repressive H3K9 methylation marks in a context-dependent manner (*Chen et al., 2012*; *Kim et al., 2017*; *Zylicz et al., 2018*). We performed an assay for transposase-accessible chromatin using sequencing (ATAC-seq) to characterize chromatin accessibility in EHMT2

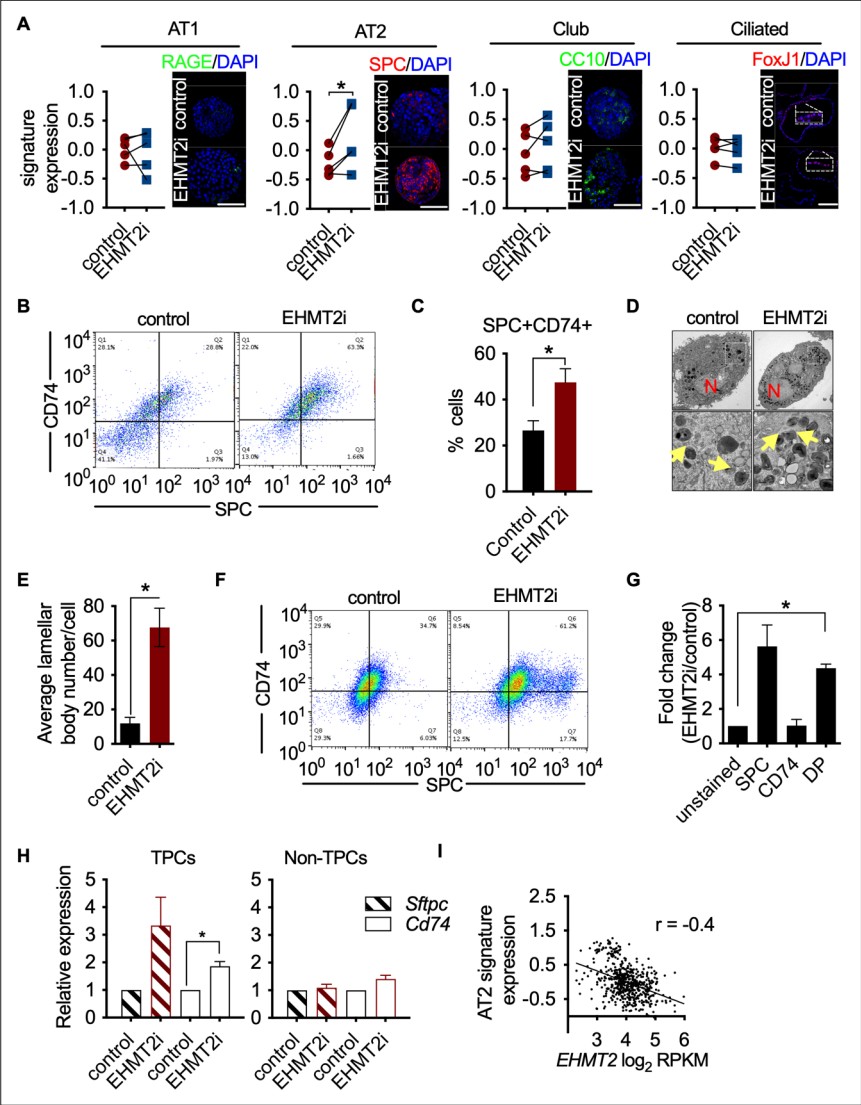

**Figure 3.** EHMT2 preserves TPC function by preventing AT2 differentiation. (**A**) Graphs show enrichment analyses of distinct alveolar cell-lineage gene signatures in transcriptomes generated from KP-derived primary tumorspheres following EHMT2 inhibition (EHMT2i) vs vehicle control (control) (n=4; mean Z-score ± SEM, two-tailed paired t-test, *p<0.05), each paired with immunofluorescence (IF) micrographs of representative canonical marker from their respective cell lineage. Scale bar 100μM (See Figure (S3E) for quantitation of IF). (**B**) Representative flow cytometry of cells derived from primary tumorspheres treated with either vehicle control (control) or EHMT2 inhibitor (EHMT2i) for 5 days and immuno-stained for alveolar type 2 (AT2) markers surfactant protein C (SPC) and CD74. (**C**) Quantification of the SPC-CD74 double positive (DP) population depicted in (**B**). (n=4; mean ± SEM, two-tailed paired t-test, *p<0.05). (**D**) Representative transmission electron microscopy (TEM) image of cells extracted from primary tumorspheres, treated as in (**B**). (Upper panel, scale bar 2 μm; lower panel, respective insets in the upper panel, scale bar 0.5 μm). (N, nucleus; yellow arrows, lamellar bodies). (**E**) Quantification of TEM in (**D**) (n=2; mean ± SEM, two-tailed paired t-test, *p<0.05). (**F**) Representative flow cytometry of tumor-propagating cells (TPCs) sorted after EHMT2 inhibitor (EHMT2i)- or vehicle control-treatment of primary tumorspheres and immuno-stained for AT2 markers SPC and CD74. (**G**) Quantification of (**F**), showing fold-change in EHMT2i/control ratio of AT2 markers SPC and CD74 (n=2; mean ± SEM, One-way ANOVA with Tukey's multiple comparison test, **p<0.005). (**H**) Relative expression of *Sftpc* and *Cd74* transcripts in EHMT2i vs control; TPC and non-TPC, respectively. (n=3; mean ± SEM, two-tailed paired t-test, *p<0.05). (**I**) Spearman's rank correlation analysis between orthogonal human AT2 gene signatures and *EHMT2* transcript in 546 human lung adenocarcinomas (LUAD). (n=546, linear regression analysis, ***p<0.0001, *r*=−0.4).

The online version of this article includes the following figure supplement(s) for figure 3:

*Figure 3 continued on next page*

*Figure 3 continued*

**Figure supplement 1.** EHMt2 inhibition reduces apoptosis and proliferation in tumorspheres.

**Figure supplement 2.** Quantification of immunostaining in control and EHMT2i-treated tumorspheres (RAGE: control, n=28; EHMT2i, n=21).

**Figure supplement 3.** Induction of surfactants following EHMT2 depletion and pharmacologic inhibition.

**Figure supplement 4.** Relative expression of alveolar type 2 (AT2) markers in tRFP-sorted tumor cells derived from primary recipients expressing either shcontrol or sh*Ehmt2* (n=3; mean ± SEM; *Sftpc, Slc34a2*, Lamp3, Cd74; two-tailed paired t-test *p<0.05).

**Figure supplement 5.** Quantification of tumor-propagating cell (TPC) fraction following EHMT2i vs control (n=19; mean ± SEM; two-tailed paired t-test *p<0.05).

**Figure supplement 6.** Spearman's rank correlation analysis between orthogonal human alveolar gene signatures and *EHMT2* transcript in 546 human lung adenocarcinomas (LUAD) (n=546, linear regression analysis p<0.0001).

inhibitor-treated, tumorsphere-derived TPCs. Surprisingly, very limited changes in chromatin accessibility were observed in TPCs upon EHMT2 inhibition (*Figure 4—figure supplement 1A,B*), indicating that EHMT2 inhibition does not induce widespread chromatin remodeling: 11 promoter and 320 non-promoter sites became more accessible following EHMT2 inhibition, whereas 3 promoter and 54 non-promoter sites were less accessible (FDR <0.05, fold-change >1.5). The promoter sites were not disproportionately enriched for any pathway, and could not account for the EHMT2-induced phenotype.

In the absence of EHMT2 i-induced chromatin accessibility changes, we investigated signaling pathways that influence alveolar cell fate decisions in TPCs. A subset of AT2 cells in normal lung act as tissue stem cells, and Wnt signaling is critical for the maintenance of their stem cell identity (*Nabhan et al., 2018*; *Zacharias et al., 2018*). We reasoned that EHMT2 could regulate Wnt signaling as a means to maintain stemness and prevent differentiation in TPCs. Consistent with this line of reasoning, EHMT2 inhibition resulted in a statistically significant increase in *Axin2* expression only within the TPC subset (*Figure 4A*), implicating a functional link between EHMT2 activity and Wnt signaling.

Since EHMT2 inhibition led to *Axin2* upregulation whilst impairing TPC self-renewal and inducing an AT2-like cell fate, we assessed whether Wnt activation alone was sufficient to achieve these outcomes. Indeed, pharmacological activation of Wnt signaling using two doses of the GSK3β inhibitor CHIR99021 (*Figure 4B* and *Figure 4C*) impaired TPC stemness as illustrated by reduced tumorsphere self-renewal, consistent with EHMT2 inhibition (*Figure 4B–F*). Moreover, corresponding Wnt pathway activation and transcriptional activation of AT2 marker genes was observed at secondary passage (*Figure 4E*), with a concordant dose-dependent increase in SPC surface expression (*Figure 4F*). To better understand the link between Wnt pathway activation and selective increase of AT2 cell lineage gene expression, we performed an unbiased analysis of transcription factor binding motifs within the promoters of cell lineage signature genes (*Treutlein et al., 2014*). We found that the *Tcf4* motif ranked eighth among 264 tested motifs for AT2 genes, while ranking much lower for other cell lineages, supporting the concept that AT2 cell fate is directly linked to a Wnt-driven signaling process (*Figure 4—figure supplement 2*). Moreover, 4 out of 7 *Tcf4*-containing AT2 genes are significantly induced in TPCs upon EHMT2 inhibition (*Figure 4—figure supplement 3*), further supporting the concept that the EHMT2 -inhibitor phenotype is Wnt-mediated. Notably, the *Tcf4*-containing AT2 genes that are induced upon EHMT2 inhibition display accessible chromatin configurations at their promoters irrespective of treatment (*Figure 4—figure supplement 4*), suggesting that these genes are poised to respond to Wnt-mediated signals and therefore would not require chromatin accessibility changes to enable gene expression (*Zacharias et al., 2018*).

## EHMT2 restrains Tcf4 gene transcription by repressing chromatin bound β-catenin through RUVBL2

Given the convincing link between inhibition of EHMT2 activity and Wnt-mediated AT2 gene expression, we sought to elucidate the mechanistic basis for Tcf4-mediated gene transcription. Since we observed only a limited change in chromatin accessibility in response to EHMT2 inhibition, we reasoned that a non-histone substrate may be controlling this process. Previous reports have shown that EHMT2 -dependent methylation of the non-histone substrate RUVBL2 (REPTIN, TIP48, and

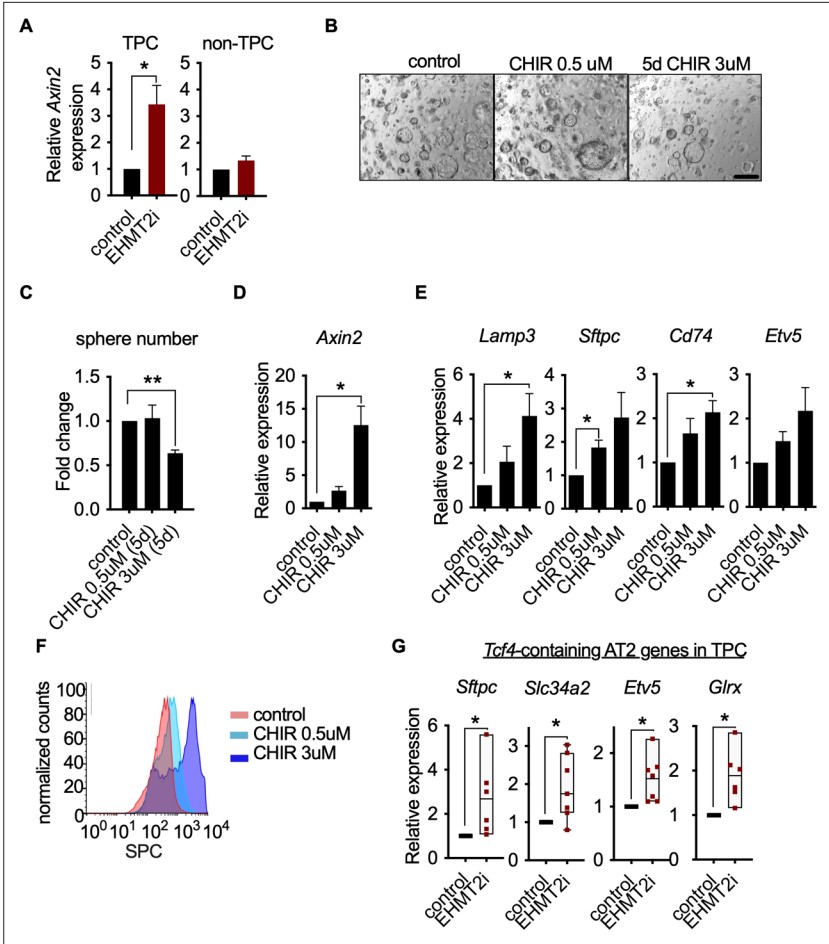

**Figure 4.** Wnt activation impairs TPC self-renewal and induces AT2 cell lineage marker expression. (**A**) Relative expression of *Axin2* transcripts in tumor-propagating cells (TPCs) and non-TPC following EHMT2 inhibition (EHMT2i) vs vehicle control (control) (n=4 mean± SEM; two-tailed paired t-test, *p<0.05). (**B**) Representative micrographs of primary tumorspheres passaged to single cells following 5 days of the GSK3b inhibitor, CHIR (5 days CHIR) at indicated doses vs vehicle control (control) and assessed for secondary sphere formation (scale bar 100 μm). (**C**) Quantification of sphere formation experiments as represented in (**b**) (n=4 mean ± SEM; two-tailed paired t-test, **p<0.005). (**D**) Relative expression of *Axin2* transcripts in primary tumorspheres (n=5 mean ± SEM, two-tailed paired t-test, *p<0.05). (**E**) Relative expression of alveolar type 2 (AT2) markers in tumorspheres (n=5 mean ± SEM, two-tailed paired t-test, *p<0.05). (**F**) Flow cytometry for surfactant protein C (SPC) in primary tumorspheres, treated with two doses of GSK3β inhibitor (CHIR) for 5 days vs vehicle control (control). (**G**) Relative expression of TCF4-containing AT2 markers in TPCs, treated with EHMT2i vs control (n=6, mean ± SEM, two-tailed paired t-test, *p<0.05).

The online version of this article includes the following figure supplement(s) for figure 4:

**Figure supplement 1.** EHMT2 inhibition does not induce broad changes in chromatin accessibility.

**Figure supplement 2.** Unbiased enrichment analysis of transcription factor (TF) binding motifs, revealing enrichment of *Tcf4* motifs in promoter regions of distal alveolar cell-lineage gene signatures vs background, depicted in red and gray color bars, respectively.

**Figure supplement 3.** Relative expression of TCF4-containing alveolar type 2 (AT2) markers in tumor-propagating cells (TPCs), treated with EHMT2i vs control (n=6, mean ± SEM,).

**Figure supplement 4.** Integrated genome viewer tracks of transposase-accessible chromatin using sequencing (ATAC-seq) generated from tumor-propagating cells (TPCs).

TIP49b) can repress HIF1α-mediated transcription (*Lee et al., 2010*). RUVBL2 has also been shown to antagonize β-catenin activity (*Bauer et al., 2000*; *Mao and Houry, 2017*). We reasoned that the non-histone substrate, RUVBL2, might function to repress β-catenin activity through a EHMT2 -mediated mechanism. First, we confirmed the RUVBL2 and β-catenin interaction in a human NSCLC cell line (*Figure 5A and B*). Immunoprecipitation of RUVBL2 showed its ability to interact with β-catenin and HIF1-α proteins in hypoxic and normoxic conditions (*Figure 5A*), consistent with previously reported results (*Lee et al., 2010*). Reciprocally, immunoprecipitation of β-catenin showed an interaction with RUVBL2. However, the RUVBL2-β-catenin interaction was reduced exclusively in the EHMT2 -inhibited context (*Figure 5B*), indicating that the RUVBL2-β-catenin interaction requires EHMT2 activity, analogous to that of the RUVBL2-HIF-1α (*Lee et al., 2010*). To better visualize the impact of EHMT2 inhibition on the RUVBL2-β-catenin interaction with the relevant TPC subset, we performed proximity ligation assay (PLA). Indeed, PLA confirmed the RUVBL2-β-catenin interaction in TPCs and demonstrated a significant loss of signal in the presence of EHMT2 I, providing additional support for the requirement of EHMT2 activity to maintain the RUVBL2-β-catenin interaction (*Figure 5C and D*).

In order to gain spatial insight into how EHMT2 inhibition impacts the relationship between EHMT2, RUVBL2, β-catenin and chromatin, we performed subcellular fractionation of tumorspheres. While EHMT2 inhibition showed loss of both EHMT2 and RUVBL2 proteins from the chromatin fraction, equal amounts of β-catenin remain chromatin-bound (*Figure 5E*). The sustained levels of chromatin-bound β-catenin following EHMT2 inhibition, suggests a critical role for EHMT2 -mediated RUVBL2 regulation that occurs at the chromatin (*Figure 5E*). Notably, the cytoplasmic fraction of controls confirms the presence of both β-catenin and RUVBL2, consistent with the cytoplasmic signal observed in the TPC PLA control (*Figure 5C–E*). Given that we established that RUVBL2 interacts with β-catenin and RUVBL2 abundance is reduced in the chromatin fraction when EHMT2 activity is inhibited, we tested whether RUVBL2 chromatin occupancy is changed specifically on *Tcf4*-containing AT2 genes in TPCs. We observed over 90% reduction in RUVBL2 promoter occupancy within *Tcf4* elements of AT2 genes *Slc34a2* and *Etv5* (*Figure 5F and G*). Taken together these results indicate that EHMT2 directly controls *Tcf4*-containing AT2 gene expression through RUVBL2-mediated repression of β-catenin transcription.

## EHMT2 controls Wnt signaling and differentiation within AT2 cells

We next explored whether EHMT2 functions similarly to regulate β-catenin activity in normal AT2 cells. Pharmacological inhibition of EHMT2 in vivo for 6 days demonstrated a robust induction of Axin2 protein expression in primary distal alveolar cells using flow cytometry (*Figure 6—figure supplement 1A,B*). To assess the impact of EHMT2 inhibitor-mediated Wnt induction on AT2 cell fate, we derived primary AT2 cells from adult murine lung and determined the ability of AT2 progenitors to form alveospheres. Ex vivo culturing of primary AT2 cells leads to the formation of alveospheres with cells resembling both AT2 and AT1 cell fates (*Barkauskas et al., 2013*; *Desai et al., 2014*). In contrast to KP tumorspheres, inhibition of EHMT2 activity did not impair ex vivo alveosphere formation; however, the resulting spheres were significantly smaller relative to controls (*Figure 6B*). During alveosphere formation and expansion, emerging cells express transcriptional and surface markers consistent with an AT2 to AT1 cell differentiation (*Barkauskas et al., 2013*; *Zacharias et al., 2018*). By analyzing expression of surface markers indicative of these fates (*Desai et al., 2014*), we observed that EHMT2 inhibition significantly reduced the proportion of AT1 cells (SPC-PDPN+) from 67.2 to 24.4% (*Figure 6—figure supplement 2*). Interestingly, while the proportion of AT2 cells (SPC +PDPN-) remained unchanged, we observed a marked increase in double positive cells (SPC +PDPN + ) from 15 to 62%. Together these data indicate that EHMT2 is required for differentiation of the AT2 progenitor pool and for complete and proper AT1 differentiation. Consistent with our previous results in TPCs, EHMT2 inhibition resulted in enhanced Wnt signaling, reflected by increased *Axin2* expression, as well as *Lgr4* and *Lgr5*, two prominent Wnt signaling pathway components (*de Lau et al., 2011*; *Figure 6D*). The observed increase in *Tcf4*-containing AT2 marker gene expression supports increased Wnt-mediated activity in primary AT2 cells as in the TPC context (*Figure 6E*). In contrast to the TPC subset, we observed significant changes in the proportion of AT1 cells, consistent with the reduced plasticity observed in AT2 cells. These data demonstrate that EHMT2-mediated regulation of Wnt signaling and cell fate is also observable in untransformed, primary AT2 cells. To test whether EHMT2 loss is similarly required for the differentiation of AT2 cells in vivo, we deleted EHMT2 in the alveolar compartment

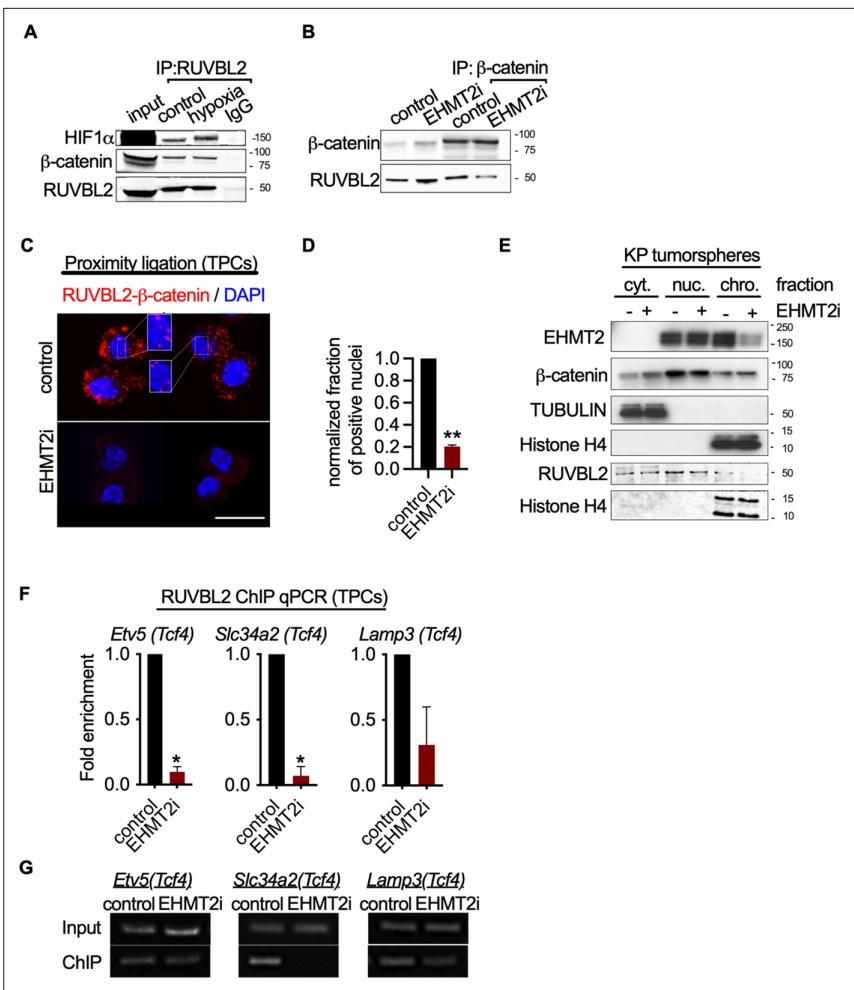

**Figure 5.** EHMT2 restrains Tcf4-mediated gene transcription by repressing chromatin bound β-catenin through RUVBL2. (**A**) Western blot demonstrating expression of HIF1-α and β-catenin in A549 lysates immunoprecipitated with RUVBL2 antibody. Cells were cultured under hypoxic conditions (1% $O_2$) vs control (ambient $O_2$). (**B**) Western blot demonstrating co-immunoprecipitation of RUVBL2 in A549 lysates co-immunoprecipitated with β-catenin antibody. Cells were treated with EHMT2 inhibitor vs control. (**C**) Proximity ligation assay in EHMT2i-treated vs vehicle-treated tumor-propagating cells (TPCs) (red, RUVBL2-β-catenin proximity ligation; blue, DAPI counterstain). Insets show a magnification of the red signal in nuclei of vehicle-treated TPCs (scale 10 μm). (**D**) Quantification of normalized nuclei with a positive signal (n=2; mean ± SEM, two-tailed paired t-test, **p<0.05). (**E**) Cytoplasmic (cyt), nuclear (nuc) and chromatin (chro) subcellular fractionation of EHMT2i-treated (as indicated) tumorspheres compared to control. Histone H4 and Tubulin are loading controls of chromatin and cytoplasmic fractions, respectively. (**F**) Chromatin immunoprecipitation using RUVBL2 antibody followed by qPCR (ChIP-qPCR) of areas flanking a *Tcf4* binding sites in promoters of the alveolar type 2 (AT2) genes *Etv5*, *Slc34a2* and *Lamp3* (n=2 mean ± SEM; *Etv5*, *Slc34a2*, two-tailed paired t-test, *p<0.05). (**G**) Representative qPCR of *Tcf4* binding motif within promoters of AT2 genes.

The online version of this article includes the following source data for figure 5:

**Source data 1.** Western blot for *Figure 5E* showing G9a (EHMT2) expression in subcellular fractionation of EHMT2i-treated tumorspheres compared to control.

**Source data 2.** Western blot for *Figure 5E* showing TUBULIN expression in middle membrane (upper band) in subcellular fractionation of EHMT2i-treated tumorspheres compared to control.

**Source data 3.** Western blot for *Figure 5E* showing histone H4 expression in the bottom membrane and beta catenin expression (upper membrane) in subcellular fractionation of EHMT2i-treated tumorspheres compared to control.

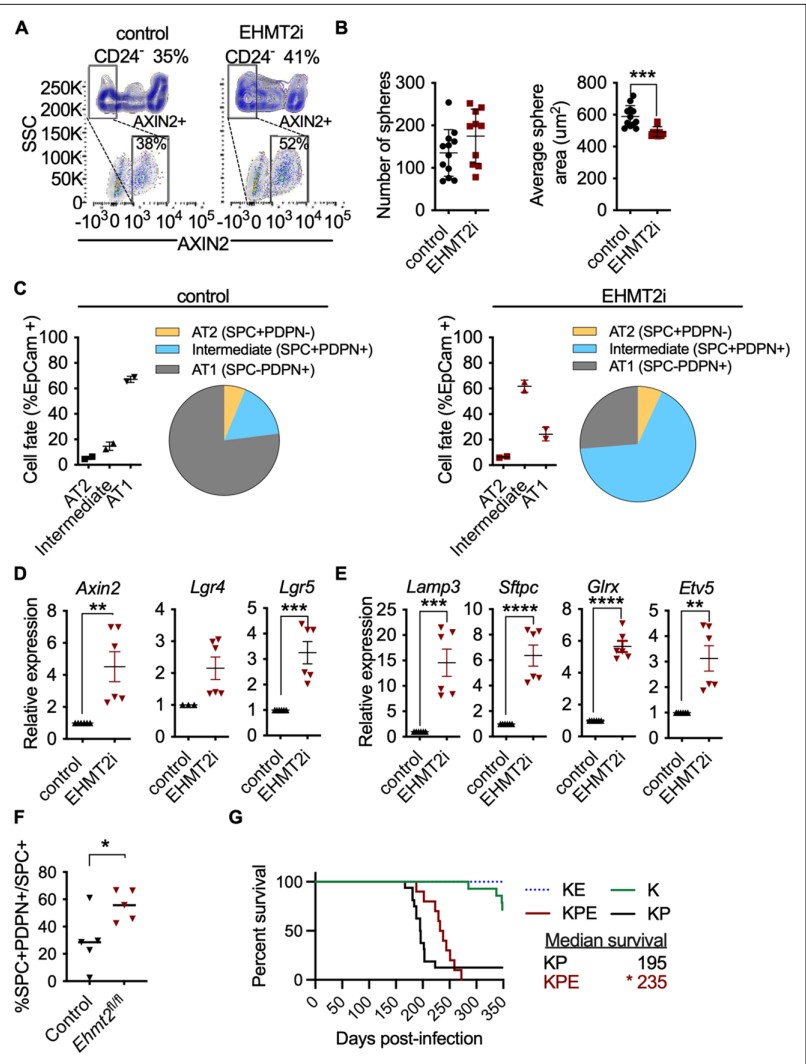

**Figure 6.** EHMT2 controls Wnt signaling and in AT2 cells and impairs tumor initiation. (**A**) Scatter and contour plots demonstrating intracellular staining of AXIN2 in CD24 negative (CD24⁻) epithelial cells sorted from EHMT2-inhibited mice compared to control (n=6 for each group). The blue contour plot shows gating on the CD24⁻ cell population. (**B**) Sphere number and size in alveospheres after treatment with EHMT2 inhibitor (EHMT2i) vs control. (n=10 ± SEM, two-tailed paired t-test, ***p<0.01). (**C**) Cell-lineage marker analysis of alveospheres, treated with either vehicle control or EHMT2i. Each panel shows a graph and a pie-chart depicting epithelial percentages of the AT1 marker podoplanin (PDPN) and the alveolar type 2 (AT2) marker surfactant protein C (SPC). (n=2 ± SEM), (**D**) Relative expression of *Axin2*, *Lgr4* and *Lgr5* transcripts (n=6–9 ± SEM, two-tailed paired t-test, **p<0.005, ***p<0.0001). (**E**) Relative expression of *tcf4*-containing AT2 transcript markers (n=6 ± SEM; two-tailed paired t-test, **p<0.05, ****p<0.0001). (**F**) Percentage of SPC +PDPN + double-positive cells out of SPC +in wildtype (control) (n=5) and *Ehmt2^{fl/fl}* (n=5) groups, 4 days post hyperoxic (75% O₂) injury. Quantitation represented as per-mouse, two-tailed paired t-test, *p<0.05 (**G**) Survival of *Kras^{G12D};Trp53* (KP) (n=16), *Kras^{G12D};Trp53;Ehmt2^{fl/f}* (KPE) (n=10), *Kras^{G12D}* (K) (n=14) and *Kras^{G12D};Ehmt2^{fl/f}* (KE) (n=9). Gehan-Breslow-Wilcoxon test, (*p<0.05).

The online version of this article includes the following figure supplement(s) for figure 6:

**Figure supplement 1.** Calibration of Axin2 antibody.

**Figure supplement 2.** Mean fluorescence intensity (MFI) of surfactant protein C (SPC) in alveolar type 2 (AT2) and intermediate alveosphere-derived cells, following EHMT2 inhibition vs control (n=2 ± SEM; Control (AT2) vs EHMT2i (AT2), two-tailed t-test, * p<0.05.

**Figure supplement 3.** Relative expression of *Ehmt2* transcript in pooled tdTomato + cells, denoting Cre exposure, from *Ehmt^{fl/fl}* vs control (n=6).

*Figure 6 continued on next page*

*Figure 6 continued*

**Figure supplement 4.** Representative image showing TdTomato-expressing cells a (yellow) and surfactant protein C (SPC)-expressing cells following 5 days of intratracheal infection with *AAV9-Cre*.

**Figure supplement 5.** Schematic representation of Hyperoxic (75% O$_2$) experiment.

**Figure supplement 6.** Representative images of surfactant protein C (SPC) + podoplanin (PDPN) + double positive cells.

**Figure supplement 7.** Decreased tumor burden in KPE mice.

by intratracheal administration of an adeno-associated virus encoding Cre (AAV9-Cre)(*Nabhan et al., 2018*). This approach allowed us to identify the EHMT2-targeted cells using a tdTomato reporter and confirm reduced EHMT2 expression from this EHMT2 -deleted population (*Figure 6—figure supplement 3*). TdT was expressed exclusively in the alveolar space and colocalized preferentially with SPC-expressing cells (*Figure 6—figure supplement 4*). To further assess how EHMT2 deletion impacts AT2 differentiation, we subsequently injured the lung using hyperoxia to promote alveolar repair (*Figure 6—figure supplement 5*). EHMT2 loss in the alveolar compartment showed a significant 2.5-fold increase in double positive (SPC + PDPN + ) cells, in just 4 days following injury consistent with our ex vivo alveosphere findings (*Figure 6—figure supplement 6F*). Studies evaluating the impact of EHMT2 loss at later stages of the tissue repair process will contribute further to our understanding of its role in epithelial cell fate decisions in the lung. All together these data demonstrate that EHMT2 functions as a cell intrinsic mechanism to directly control cell fate decisions in the context of primary, untransformed AT2 cells.

The impaired regenerative capacity of AT2 cells observed by EHMT2 loss, together with the Wnt signaling effects impelled us to examine the outcome of EHMT2 deletion in Kras-dependent tumor initiation–an event that was previously shown to require cell fate alterations in other tissue contexts (*Shibata et al., 2018*). Additionally, genetic cooperativity between Ras and Wnt signaling pathways has been reported (*Juan et al., 2014*; *Pacheco-Pinedo et al., 2011*) and linked to cell fate effects. In this case, mutant beta catenin within Scgb1A1 + cells caused a distal cell fate change and enhanced tumor formation within Scgb1A1 + expressing cells (*Pacheco-Pinedo et al., 2011*). To assess the impact on tumor initiation we deleted EHMT2 using a conditional allele of the *Ehmt2* gene (*Ehmt2* $^{fl/fl}$) together with conditional *Kras*$^{G12D}$ in the absence or presence of *Trp53* loss (KPE: *Kras*$^{lsl.G12D/wt}$: *Trp53*$^{fl/fl}$; *Ehmt2*$^{fl/fl}$ and KE: *Kras*$^{lsl.G12D/wt}$; *Ehmt2*$^{fl/fl}$) KPE mice showed a striking reduction in tumor formation and significant decrease in tumor burden in comparison to KP mice, which translated to a significant increase in overall survival (*Figure 6—figure supplement 7A,B*). Of note, the observed increase in overall survival in both KPE and KE mice was independent of p53 status. These results demonstrate that EHMT2 is crucial for Kras-mediated tumor initiation and are consistent with the requirement of EHMT2 to enable AT2 and TPC regenerative capacity and cell fate.

## Discussion

Wnt signals are critical for maintaining or acquiring facultative stem-like properties in lung tissue and distinct disease contexts. Our work reveals a cell-intrinsic mechanism of Wnt pathway activation governed by EHMT2 methyltransferase activity. We find that EHMT2 inhibition directly activates transcriptional activity of chromatin-bound β-catenin within cells, revealing an intracellular mechanism of Wnt signaling control. Interestingly, we detected RUVBL2 and β-catenin interaction in both the cytoplasm and nucleus. We currently do not completely understand the role of RUVBL2-β-catenin interaction in the cytoplasm and in which subcellular compartment this interaction is regulated. One possible explanation is that RUVBL2-β-catenin complex shuttles between the nucleus and cytoplasm compartments. In the context of lung, EHMT2 activity functions to support regenerative properties of Kras$^{G12D}$ tumors and normal AT2 cells – the predominant cell of origin of this cancer. By discovering this alternative cell-intrinsic mechanism of governing Wnt signaling in stem-like cells, we are able to manipulate cell fate decisions to predictably limit the functionality of these cells to disable tumor propagation.

The ability of AT2 stem populations to give rise to differentiated AT2 and AT1 cells is a critical feature that aids in maintaining the integrity of the distal alveolar compartment (*Nabhan et al., 2018*; *Zacharias et al., 2018*). How stemness is regulated and maintained is of considerable interest since

its dysregulation is an underlying contributor to diseases of the airway, including idiopathic pulmonary fibrosis and cancer. Wnt signaling is a master regulator of AT2 cell fate within the lung, paramount for development, homeostasis and damage repair. Regulation and tuning of Wnt signals are necessary to ensure proper repair in the alveolar compartment, but the mechanisms are context dependent. For example, β-catenin loss in AT2 cells enhances AT1 conversion, indicating Wnt signaling functionally prevents differentiation of alveolar AT2 stem cells (*Frank et al., 2016*; *Nabhan et al., 2018*). Our findings are consistent with this paradigm as functional perturbation of EHMT2 activity enhances Wnt signaling, indicating impairment of AT2 plasticity, as well as the regenerative capacity of TPCs. Of note, another report found reduced Wnt activity following EHMT2 perturbation, in three cell tumor cell lines; A549, H1299, and H1975 which invokes an APC2-mediated mechanism, albeit in a distinct cellular context from what we describe (*Zhang et al., 2018*). Indeed, the authors report that EHMT2 inhibition impairs cell growth by inhibiting Wnt. Importantly in our studies, we do not observe Wnt inhibition upon EHMT2 manipulation in either the TPC or non-TPC populations (*Figure 4A*). Additionally, the TPC population responsible for long-term regenerative capability of autochthonous tumors is not represented in established cell lines. Therefore, we view this report as a distinct context, represented by effects seen in three lung cell lines that notably all harbor mutations in ARID4A, a protein responsible for HDAC recruitment. More recent work outlines another link between EHMT2 and Wnt pathway suppression. Pal and colleagues described a role for EHMT2-mediated suppression of Wnt signaling in Rhabdomyosarcoma through activation of DKK1 (*Pal et al., 2020*). Of note, DKK1 is not expressed in AT2 cells (*Habermann et al., 2020*), thus represents a distinct mechanism of Wnt regulation from what our work describes. Although the Wnt-mediated suppression differs mechanistically from our report, this work reinforces the concept of a role for EHMT2 in suppression of Wnt signaling. Recent reports (*Nabhan et al., 2018*; *Zacharias et al., 2018*; *Zepp et al., 2017*) describe, how Wnt regulation is remarkably nuanced within the distal lung with the discovery and characterization of distinct niches that provide exogenous Wnt ligands to support neighboring stem cells. In the context of tissue damage, these interactions are likely disrupted and thereafter differentiated cells can adopt a facultative state defined by changes in Wnt signaling. Although autocrine Wnt secretion has been proposed as a mechanism to self-sustain the Wnt signaling requirements while outside of the niche, our discovery provides a means to enable cell intrinsic regulation of β-catenin transcription, providing a safe switch to maintain AT2 cell identity when a mesenchymal Wnt-providing niche, is no longer tethered to the epithelial cell.

In lung tumorigenesis, we observe that EHMT2 loss at the time of tumor initiation leads to a significant reduction in tumor formation, as well as, tumor burden resulting in significantly increased overall survival. Of note, genetic cooperativity between Ras and Wnt signaling pathways has been reported (*Juan et al., 2014*; *Pacheco-Pinedo et al., 2011*) and linked to cell fate effects. In the context of *Kras*-mutant lung tumor initiation, genetic activation of Wnt signaling using mutant beta catenin within *Scgb1A1 +* cells leads to a distal cell fate change consistent with our results. However, in contrast to our AT2-derived tumor-initiation model, mutant beta catenin expression results in enhanced tumor formation within *Scgb1A1+*expressing cells (*Pacheco-Pinedo et al., 2011*). Notably, this work differs in the cell of origin, but in that regard, it is not yet clear how distinct, alternate thresholds of Wnt signaling contribute to determining cell fate within different cell lineages in the lung. Moreover, Pacheco-Pinedo et al., used a stable form of beta catenin, whereas loss of EHMT2 activity leads to signaling of chromatin-bound beta catenin, which likely differ in both signal strength and duration. Others have reported the necessity of Wnt signaling to maintain tumors (*Juan et al., 2014*) and suggested that Wnt gradients might be critical for self-renewal, in line with the necessity for a specific Wnt threshold to support this process (*Tammela et al., 2017*). Of note, a recent report surprisingly demonstrated enhanced tumor formation when initiating tumors with mutant Kras and EHMT2 knockdown (*Rowbotham et al., 2018*), This is in direct contradiction to our findings using a genetic model to delete EHMT2 concurrent with $Kras^{G12D}$ in tumor initiation. Importantly, the shRNA seed sequence used in the Rowbotham study targets nine other target transcripts with 100% homology in addition to *Ehmt2* (*Supplementary file 1*), raising the possibility that genes other than *Ehmt2* may be implicated in the described phenotype. Notably, one of the genes targeted by this shRNA construct is *Babam1*, which is known to increase the metastatic capability of a murine Kras-mutant lung cancer-derived cell line (*Chen et al., 2015*), (*Supplementary file 1*). While in contrast to our findings, these technical differences preclude definitive conclusions from those experiments within their study.

Our work implicates EHMT2 as an important mediator of Wnt signals in order to maintain tumor propagating capacity. This cell autonomous mechanism for activating Wnt signaling may be beneficial in contexts of tumor seeding and regrowth, allowing independence from niche factors. EHMT2 activity functions similarly in primary AT2, as well as, *Kras*-transformed tumor cells, suggesting that TPCs may share functional features of their respective cell of origin. The discovery that EHMT2 functions to directly regulate Wnt signaling deepens our mechanistic understanding of EHMT2 activity and presents potential opportunities for targeting in both lung cancer, as well as other AT2-mediated lung pathologies.

# Materials and methods

**Key resources table**

| Reagent type (species) or resource | Designation | Source or reference | Identifiers | Additional information |
|---|---|---|---|---|
| Cell lines (Homo-sapiens) | A549 | ATCC | CRM-CCL-185 | Short tandem repeat profiling, SNP fingerprinting, and mycoplasma testing were used for strict quality control. |
| Cell lines (Homo-sapiens) | MRC5 | ATCC | CCL-171 | Short tandem repeat profiling, SNP fingerprinting, and mycoplasma testing were used for strict quality control. |
| Cell lines (Homo-sapiens) | RKO | ATCC | CRL-2577 | Short tandem repeat profiling, SNP fingerprinting, and mycoplasma testing were used for strict quality control. |
| Antibody | (rat monoclonal); CD45 biotin conjugated | BD Biosciences | 553078; clone 20-F11 | (1:200) |
| Antibody | (rat monoclonal);Ter119 biotin conjugated | BD Biosciences | 553672; clone Ter119 | (1:200) |
| Antibody | (goat polyclonal); clone MEC13.3; CD31 biotin conjugated | BD Biosciences | 553371 | (1:200) |
| Antibody | (rat monoclonal); CD24 PerCP-eFluor 710 | eBioscience | 46–0242 | (1:300) |
| Antibody | (rat monoclonal); EPCAM-FITC | Biolegend | 118208; Clone G8.8 | (1:20) |
| Antibody | (rat monoclonal); ITGB4-PE | Biolegend | 123602; clone 346–11 A | (1:20) |
| Antibody | (armenian hamster monoclonal); Notch1-APC | Biolegend | 130613; clone HMN1-12 | (1:80) |
| Antibody | (armenian hamster monoclonal); Notch2-APC | Biolegend | 130714; clone HMN2-35 | (1:80) |
| Antibody | (armenian hamster monoclonal); Notch3-APC | eBioescience | 17-5763-82; clone HMN3-133 | (1:80) |
| Antibody | (armenian hamster monoclonal); Notch4-APC | Biolegend | 128413; clone HMN4-14 | (1:80) |
| Antibody | (rat monoclonal); Fc-Block | BD Biosciences | 553141; clone 2.4 /G2 | (1:1000) |
| Antibody | (mouse monoclonal); CD74-BUV395 | BD Biosciences | 740274; clone In-1 | (1:25) |
| Antibody | (rabbit polyclonal); Pro-SPC | Abcam | ab170699 | (1:200) |
| Antibody | (rabbit monoclonal); G9a | Abcam | ab185050; clone EPR18894 | (1:1000) |
| Antibody | (mouse monoclonal); Podoplanin | ThermoFisher | MA5-16113; clone 8.1.1 | (1:200) |
| Antibody | (rabbit polyclonal); Alexa Fluor 488 | ThermoFisher | A-21206 | (1:500) |

*Continued on next page*

*Continued*

| Reagent type (species) or resource | Designation | Source or reference | Identifiers | Additional information |
|---|---|---|---|---|
| Antibody | (mouse monoclonal); BrdU | NeoMarkers | MS-1058-PO; clone BRD.3 | (1:200) |
| Antibody | (mouse monoclonal); Cleaved Caspase-3 | Cell signaling Technology | 9661 | (1:300) |
| Antibody | (rabbit polyclonal); CC10 | SantaCruz Biotechnology | 9772 | (1:200) |
| Antibody | (mouse monoclonal); FoxJ1 | eBioscience | 14-9965-82; clone 2A5 | (1:25) |
| Antibody | (rat monoclonal); Rage | R&D Systems | 175410; clone MAB1179 | (1:100) |
| Antibody | (mouse monoclonal); Actin | BD Biosciences | 612656; clone C4 | (1:20,000) |
| Antibody | (rabbit polyclonal); Axin2 | Abcam | 109307; clone EPR2005 | (1:100) |
| Antibody | (mouse monoclonal); H3K9me2 | Abcam | Ab1220 | (1:1000) |
| Antibody | (rabbit polyclonal); RUVBL2 | Bethyl Laboratories | A302-536 | Western |
| Antibody | (mouse monoclonal); Beta-catenin | BD Biosciences | 610153 | Western |
| Antibody | (rabbit polyclonal); H3K9me3 | Active Motif | 39161 | (1:1000) |
| Antibody | (mouse monoclonal); Histone H3 | Cell Signaling Technology | 3638; clone 96C10 | (1:1000) |
| Antibody | (rabbit polyclonal); RUVBL2 | Bethyl Laboratories | A5302-537A | Immunoprecipitation |
| Antibody | (rabbit polyclonal); Beta-catenin | ThermoFisher | 71–2700 | Immunoprecipitation |
| Chemical compound, drug | PE/Cy7 Streptavidin | Biolegend | 405206 | (1:300) |
| Chemical compound, drug | UNC0642 | Biotechne | 5132 | |
| Chemical compound, drug | CHIR99021 | Biotechne | 4423 | |
| Commercial assay, kit | Matrigel | Corning | 356231 | |
| Commercial assay, kit | Dynabeads Co-IP kit | ThermoFisher | 14–321 | |
| Commercial assay, kit | PerfeCTa | QuantaBio | 95146–005 | |
| Commercial assay, kit | True-ChIP Kit | Diagenode | C01010140 | |
| Commercial assay, kit | SAGM media | Lonza | CC-3118 | |
| Commercial assay, kit | Collagenase/Dispase | Roche | COLLD-RO | 2 ug/ml final concentration |
| Transfected construct (bacteriophage P1) | Adeno CMV-Cre | Baylor College of Medicine | Adeno CMV-Cre | Concentration $1.2 \times 10^7$ plaque-forming units |
| Transfected construct (bacteriophage P1) | Adeno-Flp-Ires-Cre | Baylor College of Medicine | Adeno-Flp-Ires-Cre | Concentration $1.2 \times 10^7$ plaque-forming units |
| Transfected construct (bacteriophage P1) | AAV9-Cre | Virovek | AAV9-Cre | 2E13 vg/ml dilution 1:60 in 60 ul vol. |
| Other | Bioruptor Pico | Diagenode | B01080010 | |
| Other | Influx cell sorter | BD Biosciences | NA | |
| Software, algorithm | Flowjo | BD Biosciences | Flowjo.com | |
| Software, algorithm | ImageJ | ImageJ | https://imagej.nih.gov/ij/ | |

| Reagent type (species) or resource | Designation | Source or reference | Identifiers | Additional information |
|---|---|---|---|---|

Primer sequences included in **Supplementary file 2**.
SPC: surfactant protein C.

## Mice

*Kras*[LSL-G12D] (**Jackson et al., 2001**), *Trp53*[flox/flox] (**Jonkers et al., 2001**), *Trp53*[frt/frt] (**Lee et al., 2012**), *Rosa26*[LSL-tdTomato] (**Madisen et al., 2010**) were licensed by Genentech Inc All animal studies were approved by the Institutional Animal Care and Use Committee at Genentech and adhere to the Guidelines for the Care and Use of Laboratory Animals (protocols 17–1217, 17–0107, and 18–1833 series). Tumors were induced in KP mice at 8–12 weeks of age using intranasal infection of AdCMV-Cre or Adeno-Flp-IRES-Cre (Baylor College of Medicine) at $1.2 \times 10^7$ plaque-forming units (PFU). In vivo treatment with EHMT2 inhibitor UNC0642 was performed by treating mice with either vehicle (60% PEG400/40% $H_2$0) or with UNC0642 10 mg/kg, IP, daily (60% PEG400/40% $H_2$0) for a duration of 6 days. Lungs were harvested and sorted for CD24[-] as previously described (**Barkauskas et al., 2013**; **McQualter et al., 2010**). For in vivo dosing experimentation of models, animals were randomized into treatment cohorts by tumor measurement, with equal numbers of male and female animals. The animals were dosed and monitored according to guidelines from the IACUC at Genentech, Inc Animals were censored for survival in an unblinded manner based on predetermined morbidity criteria.

## *Ehmt2* conditional knockout design

*Ehmt2* expression vector was constructed by introducing a Frt-PGK-em7-NEO-Frt as a selection marker. LoxP sites were introduced into the 5' and 3' homology sequences flanking the targeted exons (genomic location 34908772–34912090 and 34916244–34918676, respectively). The loxP sites, flanked exon 25–27, constituting the SET catalytic domain of *Ehmt2*. 5' and 3' extra genomic regions were used to design PCR primers to validate the targeted deletion which generates a 2.2 kb fragment upon Cre recombinase administration. Conditional gene deletion in the adult was generated in the lung upon Adeno-Cre administration. Licensing and strain availability upon request from Genentech (**Scheme 1**).

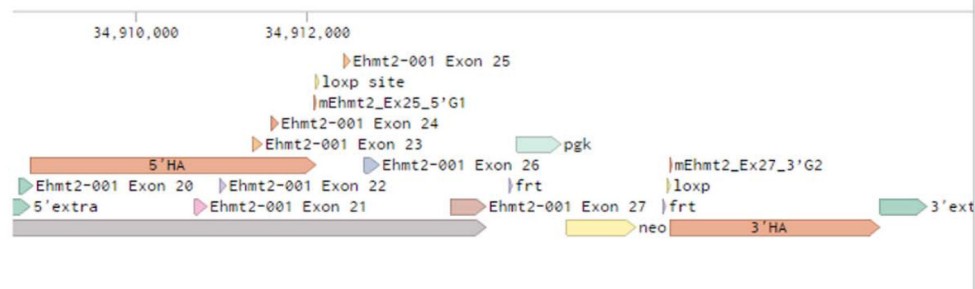

**Scheme 1.** Ehmt2 CKO-Genomic Structure.

## Orthotopic transplantation studies

For tumor formation, 8–12 week-old recipient mice were intratracheally transplanted with 25,000–45,000 inducible shRNA-carrying primary KP tumor cells per mouse. Cells were resuspended in 60 µL MEM alpha (Gibco) before transplantation and monitored using micro CT as previously described (**Zheng et al., 2013**).

## Cell isolation and tumorsphere preparation

Pooled KP tumors from 3 to 6 mice per experiment, were extracted out of the lung and completely minced with a razor blade. The material was resuspended in DMEM-F12 media containing 10% fetal bovine serum (FBS), $1 \times$ P/S and 2 µg/ml collagenase-dispase in 50 ml conical tubes and incubated for 1–1.5 hr in a 37°C incubator on a shaking platform. Digested material was sequentially filtered through 70 micron and 40 micron strainers, distributed into 15 ml conical tubes and centrifuged for 5 min at 500 g. Pellets were resuspended in hypotonic lysis buffer (15 mM $NH_4Cl$, 10 mM $KHCO_3$, and 0.1 mM EDTA) for 1–2 min, neutralized with DMEM-F12 and spun down on a 1 ml FBS cushion to remove cell debris. Final cell pellets were resuspended in PBS containing 10% FBS. For tumorsphere assays, KP tumor cells were mixed with Matrigel in tumorsphere media as previously described (*Zheng et al., 2013*). TPCs were sorted and analyzed using the influx machine (BD). Sorting was performed as previously described by *Zheng et al., 2013* with the following modifications. Immune cell lineage content was excluded by sorting with the following antibodies: biotin-conjugated CD45 (BD, 553078, 30-F11,1:200), CD31(BD, 553371, MEC13.3, 1:200), Ter119 (BD, 553672, Ter119, 1:200), thereafter biotin-conjugated antibodies were detected with phycoerythrin (PE)/Cy7 streptavidin (Biolegend, 405206: 1:300), and subsequently stained for EpCAM +FITC (Biolegend, 118208, G8.8, 1:20) and TPC markers CD24 PerCP-eFluor 710 (eBioscience, 46–0242, M1/69: 1:300), ITGB4-PE (Biolegend, 123602, 346–11 A, 1:20) and Notch1 (Biolegend, 130613, HMN1-12, 1:80), Notch2 (Biolegend, 130714, HMN-2–35, 1:80), Notch3 (eBioescience, 17-5763-82, HMN3-133, 1:80), Notch4 (Biolegend, 128413, HMN4-14, 1:80). All anti-notch antibodies are allophycocyanin (APC) conjugated and used as a pool.

## Transduction of primary KP tumor cells

shRNAs containing the following *Ehmt2* hairpins shG9.1: 5′ acagcaagtctgaagtcgaa 3′, shG9a.2: 5′ cact gtcaccgtcggcgatga 3′ were synthesized, cloned into a mirE backbone and then subsequently sub-cloned into pInducer-10 (*Fellmann et al., 2013*; *Meerbrey et al., 2011*) and transfected with packaging constructs into 293T cells using Lipofectamine 2000 (Thermo Fisher). Viral supernatants were collected 72 hr-post transfection and subsequently concentrated using ultracentrifuge at 25,000 RPM for 2 hr to generate high-titer virus. Freshly-sorted primary KP tumor cells were infected over-night using ultra-low attachment plates (corning) and subsequently transplanted or grown in Matrigel as tumorspheres.

## TPC gating strategy

Cells and doublet cells were excluded from the sort, respectively. EpCAM-Lineage scatter plot was used to gate on epithelial cells. Each TPC marker was gated in an individual scatter plot against a mock gate in the following sub-gating scheme: CD24 + cells gated from EpCAM + cells, ITGB4 + cells gated from CD24 + cells and Notch$^{HI}$ gated from ITGB4 + cells. Non-TPCs were classified as the 'non-positive' cells for each individual marker scheme and gated accordingly. All non-TPC were combined in one single tube.

## Tumorsphere and TPC assays

For EHMT2 inhibition assays in KP tumorspheres, 10,000–20,000 KP primary tumor cells from 3 to 6 mice for each biological replicate were seeded in Matrigel for 4–5 days before treatment. Each biological replicate contained n=3–4 technical replicates. Tumorspheres were then treated with either vehicle or 2 µM of the EHMT2 inhibitor Unc0642 (Bio-techne) for an additional 5–7 days. For Wnt activation assays, established tumorspheres were treated with either 0.5 µM or 3 µM CHIR99021 (Tocris) for 5 days. For EHMT2 inhibition assays in TPCs, $5x10^6$–$10 \times 10^6$ KP primary tumor cells from 6 to 8 mice per experiment were seeded per 5 ml Matrigel plug for 4–5 days before treatment and treated as described above. Cells were then extracted out from Matrigel as single cell suspensions and sorted for the TPC cell subset. TPC isolation directly from treated or untreated cultured tumorspheres was performed using the influx sorter (BD).

## Secondary sphere formation assays

For secondary sphere formation assays following EHMT2 inhibition or depletion, established KP tumorspheres were mechanically and enzymatically dissociated with 2 µg/ml collagenase/Dispase (Roche).

To obtain single-cell suspensions, dissociated KP tumorspheres were further treated with Accutase (Corning) for 5–10 min, counted and re-seeded in Matrigel in 3–4 technical replicates per experiment. Matrigel, spheres were then counted using a 10×20 eyepiece containing 0.5×0.5 mm grid.

## Alveolar and AT2-derived alveosphere preparations

AT2 derived alveosphere assays were generated by isolating Epcam$^+$/CD24$^-$ cells as previously described (*Barkauskas et al., 2013*; *McQualter et al., 2010*). Sorted cells were either intracellularly stained for Axin2 Abcam (109307, EPR2005, 1:100) or alternatively, resuspended in SAGM media (Lonza) and mixed in a 1:1 ratio with growth factor-reduced Matrigel (BD Biosciences). 100 µl of mixed cells/Matrigel suspension was placed in 24-well Transwell inserts (Falcon) and allowed to solidify. To allow alveospheres growth, $5 \times 10^4$ MRC5 human lung fibroblasts (ATCC CCL-171) were seeded in the bottom chamber supplied with 500 µl MTEC media. Media was changed every other day. EHMT2 inhibitor was replaced every 2–3 days.

## Tumorsphere immunofluorescence (IF)

KP tumorspheres were either stained in Matrigel or were paraffin-embedded, sectioned and stained as previously described (*Huber et al., 2015*). For Matrigel preparations, tumorspheres were fixed in 4% PFA for 40 min, washed 3×5 min, permeabilized with 0.5% Triton X-100 for 30 min then blocked in 4% BSA for 30–60 min. Staining was performed in blocking buffer with 0.05% Triton X-100 and washes were performed with 0.1% Triton X-100. Tumorspheres were imaged using a Leica SPE confocal microscope. The following antibodies were used for IF: BrdU (NeoMarkers, MS-1058-PO, BRD.3 1:200), cleaved caspase-3 (Cell Signaling Technology, 9661 1:300) pro-SPC (Abcam, ab170699: 1:200), CC10 (SantaCruz, 9772, 1:200) FoxJ1 (eBioscience, 14-9965-82, 2A5), RAGE (R&D, 175410, MAB1179, 1:100).

## BrdU incorporation in tumorspheres

BrdU labeling reagent (ThermoFisher) was incorporated for 3 hr. Tumorspheres were then processed according to IF procedures described in the previous section.

## Quantification of tumorsphere IF images

Images were taken using SP5 Confocal (Leica). Single projection images of 4–6 z-stack sections from at least 15–20 tumorspheres were constructed and analyzed in the Matlab software package (version R2016b by Mathworks, Natick, MA). Individual cell nuclei were segmented using regional intensity maxima and watershed thresholding on the DAPI channel, and then scored by the presence of relevant IF signal above a global intensity threshold in the area immediately surrounding each nucleus. For BrdU images, signal was calculated in each nucleus around intensity maxima. Quality control images were also created by superimposing the cell scoring mask on the raw image data.

## IP and chromatin immunoprecipitation (ChIP)

For Co-IP experiments β-catenin and RUVBL2 were pulled-down from vehicle or EHMT2 inhibitor-treated A549, using Thermo Fisher (71–2700) or Bethyl (A5302-537A) antibodies, respectively, and utilizing the Dynabeads Co-IP kit (14-321) as indicated by protocol. To detect interacting proteins, membrane was blotted for β-catenin BD (610153) or RUVBL2 Bethyl (A302-536A). ChIP experiments were performed on tumor propagating cells (TPCs) sorted directly from vehicle or EHMT2 inhibitor treated-tumorspheres using the Diagenode True-ChIP protocol (C01010140) with the following modifications. Briefly, 50 K-100K cells were crosslinked with 1% formaldehyde for 8 min, chromatin was sheared to 100–500 bp fragments using the Bioruptor Pico sonicator for 3 cycles of 30 s on/off in 0.5 ml tubes (Diagenode). Quantitation of chromatin was performed using a qubit fluorimeter where at least 10 ng of chromatin were used per experiment. Pulldown was performed with 0.5 ug RUVBL2 antibody Bethyl (A302-537A) overnight then incubated with 30 min preblocked protein A conjugated Dynabeads. Immunoprecipitated material was then washed as indicated by protocol with an additional LiCl buffer wash (0.25 M LiCl, 1% IGEPAL, 1% deoxycholic acid, 1 mM EDTA, 10 mM Tris, pH 8.1). Samples corss-links where then reversed and then samples were preamplified using PerfeCTa, QuantaBio (95146–005). The precipitated DNA and input DNA were quantified by qPCR using specific primers flanking a TCF4 binding site in the promoters of the following genes, *Etv5, Slc34a2, Lamp3*.

## Cell lineage marker analysis

Mouse cell lineage markers were obtained from *Treutlein et al., 2014* using the following selection criteria: Pearson correlation $\geq 0.5$, p-value (GBA, BH corrected)<0.05. This resulted in 30 mouse genes characteristic of AT1 cells, 28 genes for AT2 cells, 43 genes for Club cells, and 84 genes for ciliated cells. The biomaRt R package was used to map mouse genes to human orthologs, resulting in 28 human AT1 markers, 26 AT2 markers, 39 club markers, and 65 ciliated markers. For the assessment of lineage markers in human and mouse samples, expression data Z scores were first calculated for each gene across the 546 human LUAD tumors, and across the 10 mouse KP tumorsphere samples. Cell lineage signature scores were then calculated for human adenocarcinoma tumors as the average z-score expression of the human markers for each cell lineage and for mouse samples as the weighted average z-scored expression of the mouse markers with weights set to $-\log^{10}$(p-value). P-values were obtained from Table S4 (*Treutlein et al., 2014*).

## Transmission Electron microscopy (TEM)

Lung tumorspheres were first fixed in modified Karnovsky's fixative and then post-fixed in freshly prepared 1% aqueous potassium ferrocyanide-osmium tetroxide (EM Sciences, Hatfield, PA), for 2 hr followed by overnight incubation in 0.5% Uranyl acetate at $4^0$C. The samples were then dehydrated followed by propylene oxide (each step was for 15 min) and embedded in Eponate 12 (Ted Pella, Redding, CA). Ultrathin sections (80 nm) were cut with an Ultracut microtome (Leica), stained with 0.2% lead citrate and examined in a JEOL JEM-1400 TEM at 80kV. Digital imaged were captured with a GATAN Ultrascan 1000 CCD camera. For each biological replicate, 20 randomly selected cells in each treatment were manually scored and quantified for lamellar bodies. Sections were imaged at low and high magnifications, scale bars 2 μm and 0.5 μm, respectively.

## qRT-PCR analysis

RNA isolation for tumorspheres was performed using the RNAeasy Micro plus kit (Qiagen). Tumorspheres were dissociated as described. Samples were then measured using nanodrop and subsequently reverse transcribed using SuperScript III (ThermoFisher). For TPCs an additional step of pre-amplification was performed using TaqMan Preamp Master Mix (ThermoFisher). qRT-PCR was performed using fast advanced PCR Master Mix (ThermoFisher). *Hprt* and *Actin* were used as reference genes for all assays. All Taqman Assays used in the study are described in *Supplementary file 2*.

## Tumorsphere subcellular fractionation

Pooled primary KP tumorspheres (n=6) from EHMT2-treated and control Matrigel cultures were extracted, fractionated and processed to generate cytoplasmic, nuclear and chromatin fractions, using Subcellular Protein Fractionation Kit Thermo Fisher (78840), as defined by the protocol.

## Antibody validation assays

Axin2 validation was conducted using both human RKO cells and intestinal sections of *APC^loxp/loxp* vs *APC^fl/fl*; *ROSA26^CreERT2*. Briefly, RKO cells were GSK3-β (CHIR)-treated for 48 hr vs controls, harvested, 4% paraformaldehyde fixed for 10 min and subsequently washed and permeabilized with 0.25% Triton X-100. Cells were then blocked with 2.5% horse serum and stained for Axin2 Abcam (ab109307, EPR2005, 1:100) for flow cytometry. Intestinal epithelial sections were stained with Axin2 (ab109307, EPR2005, 1:25, AR; Citrate pH = 6).

## Western blot for TPCs and tumorspheres

TPCs and non-TPCs were sorted from 60 tumor-bearing mice to generate material for Western. Whole cell extracts were lysed in RIPA buffer (50 mM Tris-HCl, 150 mM NaCl, 1% Deoxycholate 0.1% SDS 1% Triton-X100) supplemented with Halt protease and phosphatase inhibitor cocktails (ThermoFisher). Lysates were quantified using pierce BCA protein assay kit (Pierce). To detect histones in tumorspheres, acid extraction method was used as previously described (*Shechter et al., 2007*). Antibodies used for Western blotting of TPCs and non-TPCs: G9a (Abcam, ab185050, EPR18894, 1:1000), Actin (BD, 612,656, C4, 1:20,000), was used as loading control. In tumorspheres; H3K9me2 (Abcam, Ab1220, 1:1000), H3K9me3 (Active Motif, 39,161 1:1000) histone H3 (cell signaling technology, 3638, 96C10, 1:1000), was used as loading control.

## RNA-sequencing data for tumorspheres

5 biological repetitions of primary tumorspheres were generated from pooled KP tumors from 6 to 10 mice per experiment. In each experiment matched KP tumorspheres were treated with either vehicle or G9a inhibitor (UNC0642). Total RNA was extracted using Qiagen RNeasy kit as per the manufacturer's protocol and quality control of RNA samples was performed to determine their quantity and quality. The concentration of RNA samples was determined using NanoDrop 8000 (Thermo Scientific) and the integrity of RNA was determined by fragment analyzer (Advanced Analytical Technologies). 0.5–100 ng of total RNA was used as an input for library preparation using TruSeq RNA Sample Preparation Kit v2 (Illumina). Size of RNA-seq libraries was confirmed using 4200 TapeStation and high sensitivity D1K screen tape (Agilent Technologies). Library concentration was determined by a qPCR-based method using library quantification kit (KAPA). The libraries were multiplexed and then sequenced on Illumina HiSeq4000 (Illumina) to generate 30 M of single end 50 base pair reads.

RNA sequencing data were analyzed with HTSeqGenie (*Pau et al., 2012*) in BioConductor (*Huber et al., 2015*) as follows: first, reads with low nucleotide qualities (70% of bases with quality <23) or rRNA and adapter contamination were removed. Reads were then aligned to the reference genome GRCm38 using GSNAP (*Wu and Nacu, 2010*). Alignments that were reported by GSNAP as 'uniquely mapping' were used for subsequent analysis. Gene expression levels were quantified as reads per kilobase of exon model per million mapped reads normalized by size factor (nRPKM), defined as number of reads aligning to a gene in a sample/(total number of uniquely mapped reads for that sample × gene length × size factor).

## Assay for transposase-accessible chromatin using sequencing (ATAC-Seq)

Cells were aliquoted into cryovials, frozen, and shipped to Epinomics, Menlo Park, CA. Cells were processed as previously described (*Buenrostro et al., 2013*). Paired-end reads were aligned to mouse reference genome GRCm38 using GSNAP (*Wu and Nacu, 2010*) version '2013-10-10', allowing a maximum of two mismatches per read sequence (parameters: '-M 2 n 10 -B 2 -I 1–pairmax-dna=1000– terminal-threshold=1000–gmap-mode=none–clip-overlap'). Duplicate reads and reads aligning to locations in the mouse genome containing substantial sequence homology to the mitochondrial chromosome or to the ENCODE consortium blacklisted regions were omitted from downstream analyses. Remaining aligned reads were used to quantify chromatin accessibility according to the ENCODE pipeline standards with minor modifications as follows. Accessible genomic locations were identified by calling peaks with MACS 2.1.0 (*Zhang et al., 2008*) using insertion-centered pseudo-fragments (73 bp–community standard) generated on the basis of the start positions of the mapped reads and a width of 250 bp. Accessible peak locations were identified as described; briefly, we called peak significance (cutoff of *P*=1e-7) on a condition-level pooled sample containing all pseudo-fragments observed in all replicates within each condition. Peaks in the pooled sample, independently identified as significant (cutoff *P*=1e-5) in two or more of the constituent biological replicates were retained, using the union of all condition-level reproducible peaks to form the atlas. (https://www.encodeproject.org/atac-seq/). The atlas consisted of 184,032 peaks with a median width of 266 bp (ranging from 250 bp to 1531 bp). 20.4% of peaks were located in promoter regions, 32.6% in intergenic regions, and 43.3% in introns.

Chromatin accessibility within each peak for each replicate was quantified as the number of pseudo-fragments overlapping a peak and normalized these estimates using the TMM method (*Robinson and Oshlack, 2010*). Differentially accessible peaks between control and EHMT2 -inhibited samples were identified using the framework of a linear model, accounting for TPC and non-TPC and implemented with the edgeR R package. Significant differences in chromatin accessibility levels within a peak between EHMT2 i-treated and control-treated samples was set to log2-fold change >1.5 and FDR <0.05. The fold-change of ATAC-seq peaks were used as input for gene set enrichment analysis using the Hallmark MsigDB gene set collection, v6.1 (*Liberzon et al., 2015*).The integrative genomics viewer (IGV) was used for visualization of ATAC-seq peaks near genes of interest (*Robinson et al., 2011*). All source code and sequencing data are available at https://github.com/anneleendaemen/G9a.CellIdentity.Lung; *Pribluda, 2020*.

## Cancer genome atlas (TCGA) RNA-sequencing data analysis

RNA-sequencing (RNAseq) data for 546 LUAD tumors from TCGA (*Cancer Genome Atlas Research, 2014*) were obtained from the National Cancer Institute Genomic Data Commons (https://gdc.cancer.gov). We employed the same approach for RNAseq data processing and quantification, with human reference genome GRCh38.

## Motif enrichment near lineage markers

For each cell lineage, function findMotifs.pl from HOMER 4.7 (*Heinz et al., 2010*) was used to identify enriched motifs near cell lineage markers from *Treutlein et al., 2014*. Motif enrichment was performed with promoter regions defined from –2000 bp to 500 bp relative to TSS, and with the background sets defined as the full genome. Locations of the TCF4 motif near cell lineage markers (from –2000 bp to 500 bp relative to TSS) were determined using findMotifs.pl from HOMER 4.7. The TCF4 motif file was obtained from the HomerMotifDB (http://homer.ucsd.edu/homer/motif/HomerMotifDB/homer-Results.html).

## Acknowledgements

We appreciate the support of all the Junttila lab members. We would like to thank Elaine Storm and the De Sauvage lab for providing critical reagents. We are grateful to the sequencing lab for their superb work. We also received technical support from the in-house genotyping and murine reproductive technology core groups, especially Tiffany Yuan. We would like to acknowledge the laboratory animal research core for facilitating the support needed to enable in vivo experimentation, as well as the center for advanced light microscopy for providing help with immunofluorescence quantitation and all FACS lab members, especially Jonathan Paw and George Tweet who were instrumental with devising critical sorting strategies.

## Additional information

### Competing interests

Ariel Pribluda, Anneleen Daemen, Anthony Nelson Lima, Erica L Jackson, Melissa R Junttila: was an employee of Genentech when the work was performed and may hold stock. Xi Wang, Marc Hafner, Chungkee Poon, Zora Modrusan, Anand Kumar Katakam, Oded Foreman, Jefferey Eastham, Jefferey Hung, Benjamin Haley, Julia T Garcia: holds shares in the company.

### Funding

No external funding was received for this work.

### Author contributions

Ariel Pribluda, Conceptualization, Formal analysis, Investigation, Methodology, Writing – original draft, Writing – review and editing; Anneleen Daemen, Marc Hafner, Formal analysis, Investigation, Methodology; Anthony Nelson Lima, Oded Foreman, Julia T Garcia, Investigation; Xi Wang, Investigation, Methodology; Chungkee Poon, Methodology; Zora Modrusan, Benjamin Haley, Resources, Methodology; Anand Kumar Katakam, Formal analysis, Methodology; Jefferey Eastham, Jefferey Hung, Formal analysis; Erica L Jackson, Supervision, Writing – review and editing; Melissa R Junttila, Conceptualization, Supervision, Writing – original draft, Writing – review and editing

### Author ORCIDs

Ariel Pribluda  http://orcid.org/0000-0003-2817-2827
Anneleen Daemen  http://orcid.org/0000-0001-6287-7105
Marc Hafner  http://orcid.org/0000-0003-1337-7598
Erica L Jackson  http://orcid.org/0000-0002-7100-8021
Melissa R Junttila  http://orcid.org/0000-0003-3538-1192

## Ethics

All animal studies were approved by the Institutional Animal Care and Use Committee at Genentech and adhere to the Guidelines for the Care and Use of Laboratory Animals (protocols 17-1217, 17-0107 and 18-1833 series).

## Decision letter and Author response

Decision letter https://doi.org/10.7554/eLife.57648.sa1
Author response https://doi.org/10.7554/eLife.57648.sa2

# Additional files

## Supplementary files

• Supplementary file 1. Summary of transcripts with 100% sequence identify to shEhmt2a seed sequence reported in *Rowbotham et al., 2018*. The seed for sh*Ehmt2* was queried in NCBI BLAST and the resulting query results are shown.

• Supplementary file 2. Primer sequences.

• MDAR checklist

## Data availability

All source code used in this study has been made available in the R computer language, in a fully documented software and data package. This package is freely available under the Creative Commons 3.0 license and can be downloaded from https://github.com/anneleendaemen/G9a.CellIdentity.Lung, (copy archived at swh:1:rev:207a6bd1022c199eaae3ef0d2fcc9e4edcbb3c69).

The following previously published datasets were used:

| Author(s) | Year | Dataset title | Dataset URL | Database and Identifier |
| --- | --- | --- | --- | --- |
| Treutlein | 2014 | High throughput quantitative whole transcriptome analysis of distal mouse lung epithelial cells from various developmental stages (E14.5, E16.5, E18.5 and adult) | https://www.ncbi.nlm.nih.gov/geo/query/acc.cgi?acc=GSE52583 | NCBI Gene Expression Omnibus, GSE52583 |
| The cancer genome atlas | 2014 | Comprehensive molecular profiling of lung adenocarcinoma | https://tcga-data.nci.nih.gov/docs/publications/luad_2014/ | tcga-data, luad_2014 |

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
