## [Editor Report]

This is generally a well-designed and carefully conducted study that is likely to be of interest to many both inside and outside of the field of lung tumorigenesis and normal lung development and tissue homeostasis.

---

## [Decision Letter]

**Decision letter after peer review:**

Thank you for submitting your article "G9a methyltransferase governs cell identity in the lung and is required for KRAS G12D tumor development and propagation" for consideration by *eLife*. Your article has been reviewed by 3 peer reviewers, including Martin McMahon as the Reviewing Editor and Reviewer #1, and the evaluation has been overseen by Edward Morrisey as the Senior Editor.

The reviewers have discussed the reviews with one another and the Reviewing Editor has drafted this decision to help you prepare a revised submission.

This is generally a well-designed and carefully conducted study that is likely to be of interest to many both inside and outside of the field of lung tumorigenesis and normal lung development and tissue homeostasis. However, there are substantial conceptual issues and issues of data interpretation that the reviewers have raised regarding the research described in this manuscript. Although it may not be feasible for the authors to address every critique listed below experimentally, the authors should attempt to address the reviewers' critiques. Some of the key issues that were raised include:

1. The results on tumor inhibition in vivo and proliferation of tumorspheres in vitro are not entirely surprising, given previously published literature by Zhang et al. (Molecular Cancer 17: 153 (2018)) showing that UNC0642 and/or shEHMT2 inhibit multiple pathways, including WNT signaling in cancer cell lines in vitro and in xenografts. The inhibitory effect on WNT signaling was ascribed to effects on the expression of APC2 resulting in increased destruction of β-catenin, a paper that authors of this manuscript do not reference. Rather, Pribluda et al. propose a model in which inhibition of G9a/EHMT2 activity in AT2 "stem cells" and in "tumor propagating cells" leads to increased WNT activity in a subset of AT2 cells and tumor cells as measured by Axin2 expression. In the case of AT2 cells this WNT pathway activation is proposed to drive the terminal differentiation of cells into a fully mature, non-proliferative AT2 cell. The authors argue that this loss of self-renewal and the potential to differentiate into AT1 cells limits tumorigenesis driven by the combination of mutationally-activated KRAS(G12D) and silencing of TP53.

2. Others have shown that either pharmacologic or genetic inhibition of WNT signaling has striking inhibitory effects on lung tumorigenesis driven by the RAS/BRAF pathway. In that context, the authors should consider the results presented in Juan et al: "Diminished WNT->β-catenin->c-MYC signaling is a barrier for malignant progression of BRAFV600E-induced lung tumors" (Genes Dev. 2014 28: 561-75). In addition, Pacheco-Pineda et al., "Wnt/B-catenin signaling accelerates mouse lung tumorigenesis by imposing an embryonic distal progenitor phenotype on lung epithelium" (J Clin Invest 121: 1935-1945) further supports the hypothesis that elevated WNT signaling promotes lung tumorigenesis. This research clearly indicates that an insufficiency of WNT signaling is inhibitory to lung tumorigenesis, which stands in contrast to the authors contention that sustained WNT signaling promotes differentiation of TPC cells to a more differentiated AT2 phenotype thereby suppressing lung tumorigenesis.

Major comments:

1. Throughout the manuscript, the authors manipulate their genes/pathways of interest largely in one direction, by inhibiting/depleting G9a, or by stimulating WNT signaling, without looking at the effects of over-expressing or rescuing G9a, or inhibiting WNT signaling. Their findings would be considerably strengthened if they could show that overexpressing G9a and inhibiting WNT signaling produced the reverse outcomes.

2. In the authors' tumorsphere assays, they have not ruled out the effects of G9a inhibition on cell proliferation and/or apoptosis, or assessed the proportion of CD24+ ITGB4+ NOTCH(hi) cells by FACs. These experiments are absent for both the in vitro and transplant experiments. These findings contradict previous literature that suggests that G9a is lost in a different TPC population and G9a inhibition increases another TPC marker, SCA-1 (Rowbotham et al. 2018). The authors should explain if these populations/markers are affected in their experiments and address any differences that they have observed.

3. It is surprising that the increase in CD74 (Difference in SP-C is not significant?) is only restricted to the TPCs and is only ~2 fold. In Figure 3, Supplement 5 the data indicates that TPCs make up ~0.12% of the sphere culture. If this was the case the increase in SPC in the whole sphere culture would be insignificant. However, in Figure 3A SPC levels are significantly higher in the G9a inhibited samples and appear widely distributed across the tumorsphere suggesting that many cells and not just the 0.12-0.02% that are TPCs have increased SPC. How do the authors account for this observation?

4. It is perhaps surprising that the authors did find many changes in chromatin accessibility in their inhibited tumorspheres, whence they conclude that G9a inhibition does not alter chromatin in their cultures. In Figure 1E however they clearly show that levels of the chromatin modifications H3K9me2 and H3K9me3 are significantly reduced suggesting that the chromatin has been altered. The authors should account for this and rule out technical problems with their ATAC-Seq data. Indeed, although the data on β-catenin⌠RUVBL2 is interesting, it remains unclear whether this interaction is indeed brokered by changes in RUVBL2 lysine methylation.

5. In the distal lung, WNT responsive Axin2+ AT2 cells have been demonstrated to act as progenitors which can regenerate a substantial fraction of the alveolar epithelium (Frank et al. 2016, Zacharias et al. 2018, Nabhan et al. 2018). Zacharias et al. suggest that SP-C+/Axin2+ cells are an alveolar progenitor distinct from AT2 cells and Nabhan et al. suggest that Axin2+ AT2 cells have stem cell activity. The authors are suggesting that in the context of a KRAS(G12D)-driven lung adenocarcinoma, WNT signaling and Axin2 expression have the opposite effect and oppose the stem cell activity of the TPCs. As this is in direct contrast to a study using the same mouse model which shows that WNT signaling supports the self-renewal and tumor propagation of TPCs, and inhibiting WNT signaling from the niche impairs tumor growth and progression (Tammela et al. 2017), the authors should test their mechanism in vivo or explain the discrepancy.

6. The authors show the RUVBL2 and β-Catenin interaction is altered by G9a inhibition in a human cell line. Can the authors also show this interaction in their tumorsphere cultures or at least with a mouse KRAS(G12D)/TP53(Null) cell line for relevance to the rest of their results?

7. In Figure 6 the authors should use a pharmacodynamic biomarker to show that the G9a inhibitor reaches the epithelial cells of the lungs and is actively inhibiting di- or tri-methylation of H3K9. The results are not consistent with the conclusion that AT2 stemness has been altered by G9a inhibition, as this has not been directed tested. The data from Figure 6B showing no difference in sphere forming ability suggests against this interpretation. The authors should passage their G9a inhibited alveolar spheres and test this directly. As the proportion of AT2 cells between the two cultures does not appear to be altered we might not expect to see a significant difference. The authors should perform statistical tests to show that the differences between cells with AT2 and AT1 markers are significantly different, even though they appear large. The results seem to suggest not that AT2 stemness is lost, but that the differentiation to AT1 cells is impaired, leading to a majority intermediate type cell (SP-C+/PDPN+) instead of a majority of AT1 cells. This would be consistent with existing literature showing that activating WNT signaling with the CHIR GSK3-β inhibitor increases the proportion of AT2 cells in culture whilst inhibiting WNT signaling increases the proportion of AT1 cells (Zacharias et al. 2018).

8. The authors administer Cre adenovirus to a G9a(fl/fl) reporter mouse to further study the effects of G9a loss in vivo. However, the authors do not describe this allele in their Materials and Methods nor do they cite an original publication of such an allele/mouse. (Perhaps this is the relevant citation: Lehnertz et al., 2010 Activating and inhibitory functions for the histone lysine methyltransferase G9a in T helper cell differentiation and function. J Exp Med. 207: 915-922). However, if this mouse is not published the authors should provide additional information on the provenance of this allele/mouse. The authors should demonstrate that G9a expression is successfully silenced in this tumor model, or if the mechanism they propose is active in vivo, whereby G9a deletion in tumors would be presumed to establish continuous WNT signaling to maintain AT2 differentiation and either prevent tumor initiation or retard tumor growth.

9. The statistical analysis of the SPC+/PDPD+ cells following hyperoxic injury should be reassessed. The >10,000 cells the authors enumerate are not independent of each other and should not be analyzed by a t-test in this manner. An n>10,000 is too high for a t-test and will almost always result in p<0.05, rendering the statistical analysis meaningless. A more relevant way to analyze this data is to average the % SPC+/PDPN+ cells per mouse and then perform a statistical test on these values (n=6), assuming each mouse was independent of the others. A t-test would be appropriate if the data are normally distributed, and if not, a non-parametric ranked test such as Mann-Whitney.

10. If the mechanism suggested by the authors is correct, then they should show that G9a is deleted in the alveolar cells that become SPC+/PDPD+ after hyperoxic injury and not in the cells that escape G9a deletion. Whilst the authors do present evidence that alveolar cell fate appears to be affected by G9a, the 'stemness' of the AT2 cells has not been measured either directly (by lineage tracing their progeny and quantifying their contribution to the new epithelia) or indirectly by studying the dynamics (i.e. timing) of regeneration following injury in G9a(fl/fl) vs normal mice.

11. Throughout the manuscript the authors refer to AT2 "cell fate identity" and to the "trans-differentiation" of AT2 to AT1 cells. This is quite confusing since there has been important work on the role of transcription factors such as *SOX2* and NKX2.1 on determining the "identity" of epithelial cells in NSCLCs (e.g. Tata et al. 2018 Dev Cell 44: 679-693 and Snyder et al. 2013 Mol. Cell. 50: 185-99). In this previous research a change in identify involves true switching of fate from lung to non-lung (e.g. intestinal or esophageal) cell types. Consequently, it is unclear whether the differentiation of AT2 cells to AT1 cells should be considered "trans-differentiation" or loss of "cell fate identify" but rather the normal behavior of AT2 cells in homeostasis and repair of the adult lung.

12. The authors should consider the confusing fact that two papers that describe active WNT signaling in a subset of AT2 "stem cells" in the normal mouse lung (Nabhan et al. 2018 and Zacharias et al. 2018) report very different proportions of these cells (~3% vs. ~30%). Indeed, the authors assume that there is a fixed subset of AT2 stem-like cells and that these are maintained in a "plastic" self-renewing state by the correct level of WNT signaling. Perhaps AT2 cells can oscillate between states in which they are more or less differentiated, and that it takes longer for the fully differentiated cells packed with lamellar bodies and associated secretory machinery to start to divide and/or give rise to AT1 cells? Indeed, there is little evidence in vivo that KRAS-driven tumors can only arise in Axin2+ AT2 "stem cells", albeit that WNT signaling is essential for both KRAS and BRAF-driven lung tumorigenesis.

13. To support their model the current authors should consider testing whether their tumorspheres contain both AT2 and AT1 cells and whether treatment with UNC0642 specifically reduces differentiation of AT1 cells and does not affect other pathways regulating cell proliferation and survival, as proposed by Zhang et al. The experiments in which Pribluda et al. test the effect of inhibiting EMHT2 on primary mouse AT2 cells in organoids (alveolosphere) culture are confusing. The authors state, "During alveosphere formation and expansion, emerging cells express transcriptional and surface markers consistent with an AT2 to AT1 cell fate transition". A "cell fate transition" was not ascribed to the normal differentiation of AT2 to AT1 cells by the original authors. The fact that the treated spheres are smaller than normal may reflect reduced cell proliferation – however, this was not assayed and the morphology of the spheres and immunohistochemistry is not shown. The authors state "Interestingly, while the proportion of SPC+/PDPN- AT2 cells (SPC+/PDPN-) remained unchanged, we observed a marked increase in double positive cells (SPC+/PDPN+) from 15% to 62%. Together these data indicate that G9a is required for differentiation of the AT2 progenitor pool and for complete and proper AT1 cell fate transition". However, the significance of dual positive cells in vivo or in vitro is not known and it is unclear how the lack of change in the proportion of AT2 cells can be interpreted as reduced differentiation of AT2 cells. Finally, it should be noted that neither LGR5 nor LGR6 are not normally expressed in lung alveolar epithelial cells in vivo – rather expression is confined to mesenchymal cells.

[Editors' note: further revisions were suggested prior to acceptance, as described below.]

Thank you for resubmitting your work entitled "G9a methyltransferase governs cell identity in the lung and is required for KRAS G12D tumor development and propagation" for further consideration by *eLife*. Your revised article has been evaluated by Edward Morrisey (Senior Editor) and a Reviewing Editor.

The manuscript has been improved but there are some remaining issues that need to be addressed, as outlined below. Indeed, as you will see from the reviews below, two of the three reviewers are willing to accept the manuscript, whereas the third reviewer has raised a number of additional concerns, not the least of which is the fact that only one of the shEHMT2 hairpins reduces sphere formation despite the fact both hairpins elicit strong reduction of EHMT2 expression. Hence, all three reviewers are concerned about the possibility of off-target effects. Hence, in order to satisfy all three reviewers, and to allow us to accept your manuscript, we ask that:

1. You address the comments raised by Reviewer #2 on a point-by-point basis.

2. That you more explicitly state in the text a concern regarding the issue of the shEHMT2 hairpins.

3. That you address R3's comments regarding citations that document expression of LGR5 in the epithelial compartment.

At this point the BRE, acting in consultation with the *eLife* Senior Editor overseeing this submission, will assess your response and make a final decision on your manuscript recognizing that, with your departure from Genentech, it is likely not possible for additional experiments to be conducted on this project.

*Reviewer #1 (General assessment and major comments):*

The authors present a substantially revised manuscript and an accompanying thorough and conscientious response to the reviewers' critiques that, in my opinion, addresses the main concerns that were expressed in the original review. While there remain some important unresolved questions, this is more the hallmark of good science and does not reflect a deficiency in the revised manuscript. Moreover, the authors have further elaborated on the generation of new alleles of Ehmt2. Consequently, I am happy to recommend that the revised manuscript be accepted for publication in eLife.

*Reviewer #2 (Recommendations for the authors):*

Pribula et al. have revised their manuscript "EHMT2 methyltransferase governs cell identity in the lung and is required for KRASG12D tumor development and propagation. In a lengthy response to the first round of reviews, the authors make some changes to improve their manuscript, particularly with regards to the interpretation of their data. However, they do not appear to have attempted any of the experiments suggested by the reviewers to improve their manuscript, particularly with regard to testing the effects of G9a inhibition/knockdown on the Sca-1+ population of KP tumors in their systems, which has been published to interact differently with G9a loss than the authors report, and connecting the mechanism established in A549 cells to their phenotypes in tumors and alveospheres.

Furthermore, in response to a minor point, the authors have highlighted a significant problem with their manuscript that was overlooked in the previous round of revisions. In Figure 1I it was noted that only one hairpin was denoted as significantly reducing tumorsphere formation. Initially assumed to be a mistake, the authors clarify that indeed only one hairpin significantly reduced tumorsphere formation. This leaves only one valid hairpin showing the same phenotype as G9a inhibition and as shRNA's are notorious for off-target effects, it cannot be concluded from just one hairpin that effects were due to observed reduction in EHMT2 mRNA.

This concern carries over into figure 2 where inducible hairpins are used to study the effects of EHMT2 knockdown on transplanted tumorigenesis in vivo. In the description of these experiments, it is not clear if the same two hairpins were used, and if both hairpins independently produced the same phenotype. Indeed, from the legend of Figure 2E, it appears that only one hairpin was used (shEHTM2 2.1). All shRNA hairpin experiments must be completed with at least two, and ideally 3+ independent hairpins to ensure that the phenotypes observed are not due to off-target effects. The sequence of the hairpins used does also not appear to be in the manuscript. Because these are serious concerns regarding experiments key to the manuscripts claim, we must conclude that the manuscript is still not suitable for publication in it's current state.

Specific points:

In response to point 1. the authors note that in discussing Zhang et al. 2018, and the results that UNC0642 and shEHMT2 inhibited wnt signaling this study relied on three human cancer cell lines, A549, H1299 and H1975. They argue that the results of this paper are context dependent on these cell lines, and not applicable to their primarily mouse GEMM studies. However, the biochemical experiments linking G9a to Wnt regulation in Pribula et al. are performed in A549 cells and not in primary GEMM tumors, organoids or AT2 cells (For obvious practical reasons). As the authors establish their G9a-wnt mechanism in this same cell line where g9a inhibition was shown to suppress wnt signaling, this should be addressed. If A549 cells are such a different biological context to their GEMM model (ARID4A mutations etc), then can they expect that their mechanism found in A549 cells carries through to GEMM tumors but the inverse finding on wnt signaling activation does not? Do the authors observe the same results with G9ai in A549 cells as Zhang et al.?

2. The authors make a complex argument that G9a regulates a precise level of wnt activity, when G9a is inhibited an increase in wnt activity drives tumors cells to a differentiated AT2 cell state. This idea of a distinct Wnt threshold is intriguing. This concept and the context of pacheo-Pinedo et al. and Juan et al. should be addressed much earlier in the manuscript when wnt signaling is first introduced and not only in the discussion. The Wnt-threshold concept suggests that either activating or inhibiting wnt could both be deleterious or supportive to lung tumors, depending on context (i.e cell of origin) which would make clinical use of wnt inhibitors/activators a challenging prospect.

Comments:

The authors regret that they were unable to experimentally inhibit wnt signaling, arguing that because in their model, because G9a functions downstream of where most wnt inhibitors act, e.g. on ligand interactions and the B-catenin complex, and because they argue that only suprabasal wnt activity leads to the observed phenotype. If this is interpretation is correct, it would actually present an elegant method to test their model, as wnt inhibition (very easy to technically administer and can be carefully titred to the correct level) should therefore not be able to rescue the effects of G9a inhibition in their system. We believe that this experiment should be completed and reported as it is simple and would greatly strengthen their claims.

2. The authors highlight some tumorsphere data showing that both proliferation and apoptosis are reduced. The purpose of asking for these data was to confirm if the effects of G9a inhibition were truly on stemness which leads to more organoids forming, or because G9a inhibition might reduce proliferation or increase apoptosis of the tumor cells. The results highlighted from figure S 3-1 show that whilst apoptosis is down, proliferation is also strongly reduced. Therefore are the results from G9ai on tumor growth, formation etc due to changes in proliferation rather than AT2 cell differentiation? Or do the authors consider that these are indistinguishable?

Whilst the discussion of the different TPC populations described is appreciated and adds important context, noting that they have different abundances. The question of whether the other TPC marker, Sca-1 and the Sca-1+ fraction is affected in their experiments remains unaddressed when a simple QPCR for Sca-1 in any of their tumor systems could help to address this.

3. We appreciate the reviewer's explanation, highlighting the difference between the in vivo and tumorsphere results. To incorporate this explanation into the text, the authors should mention that the broadly increased SPC expression observed in the cultures is likely from progenitors and differentiated tumor cells whilst in vivo only TPCs show an increase in genes associated with AT2 differentiation.

4. The authors helpfully remind the reviewer that chromatin accessibility was measured in TPCs and not tumorspheres, where they demonstrate that H3K9me2 and H3K9me3 are, as expected, reduced by G9a inhibition. Although they find very few changes in chromatin accessibility in TPCs, the method they describe is appropriate and if carried out correctly should find differences if they are there. This explanation seems to suggest that chromatin modifications would not be changed in TPCs and differences are restricted to tumorspheres "a more diverse population including both TPCs and non-TPC progeny". The authors would seem to be suggesting, therefore, that unlike tumorspheres in culture, H3K9me2 and H3K9me3 are not reduced by G9a inhibition in TPCs. This would be a surprising result, and at the very least should be investigated by the authors.

5. In the explanation provided the authors refine their explanation of the effects of G9a and WNT signaling on TPCs, to align their conclusions with the existing literature. As mentioned in earlier rebuttal points, they interpret their data to suggest that G9a restriction of Wnt signaling is supportive of TPC function, and if G9a is lost or inhibited then excess wnt signaling can lead to loss of TPC function. This represents an interesting finding, although the authors perhaps overinterpret their findings when they suggest that G9a regulates Wnt to a distinct threshold (author's emphasis).

With the authors data and the existing literature suggest that either too little, or now too much Wnt signaling can be deleterious to TPCs and AT2 cells. Whether there is a distinct threshold of wnt signaling for TPCs/AT2 cells to maintain proper function, and more importantly, what this threshold might be is interesting to speculate about, but goes beyond what the author's data show. It would take a considerable amount of experimentation to demonstrate that such a distinct threshold exists, more than could be reasonably expected in a revision of this manuscript. The authors should amend the description of their results and the discussion appropriately.

6. It is unfortunate that the reviewers were not able to confirm the RUVBL2 and B-Catenin interaction in tumorspheres or TPCs, and that mouse cell lines do not behave like TPCs in order for them to be used as a model. This raises the question then, if the A549 cells used for the mechanistic experiments contain "the relevant population to progeny relationship found in de novo kp tumors". Do A549 cells behave similarly to TPCs in organoid culture in response to G9a inhibition? This is important for the connection of the proposed mechanism to the phenotypes observed in the mouse. If the mechanistic relationship cannot be tested in the phenotypic model(s), then the phenotype should be confirmed in the mechanistic model, otherwise we cannot connect the mechanism to the phenotype even if they are 'operating in the same direction' (positively corelating). As mentioned earlier in this discussion, Zhang et al. found that Wnt signaling was inhibited by G9a inhibition and EHMT2 shRNA in cultures and xenografts of human cells including A549 cells. Whilst this study should not be considered the last word on this topic, it highlights the importance for the authors to test to see if their phenotype can be reproduced in this system used to elucidate their proposed mechanism.

Whilst the authors have convincingly shown that Wnt signaling is modulated upwards in response to G9a inhibition in tumors and AT2 cells, a stronger connection to the proposed mechanisms is required for publication.

7. It is helpful that the authors provide pharmacokinetic data showing consistent plasma concentration of their inhibitor following. We take it that the G9a inhibitor is highly selective over other histone methyltransferases which has been amply demonstrated in the original literature. Whilst direct evidence of G9a inhibition in vivo would (through histone methylation) would be greatly convincing, we appreciate that confirmation of the main phenotypes of increases of wnt signaling and AT2 gene expression through genetic deletion makes it likely that G9a inhibitor is functioning as expected in vivo. In the absence of confirming whether G9a inhibition reduces the stem cell capacity by serial passaging of inhibited AT2 cell cultures, the text changes made by the authors is appropriate.

8. The author's test if there are changes in the proportion of AT1 markers in their tumorsphere assays with G9a inhibition, as they see with their alveolar spheres. Here they do not see a significant change AT1 cell markers, highlight differences between how tumor cells and AT2 cells respond to G9a inhibition, which should be commented on in the discussion.

The authors have already addressed wording changes to the descriptions of their alveolar sphere cultures to reflect that AT2 to AT1 is the normal differentiation process and not an example of trans-differentiation.

Whilst the authors' results are consistent with the explanation that a change in AT2 to AT1 differentiation, the small size of the alveospheres does suggest that changes in proliferation could also contribute to these differences, and really should be explored by the authors. The immunohistochemistry would also be extremely informative as well, what to the AT1/AT2 cells look like?

The authors point out that it is not possible to perform secondary transplantations with their inducible due to the re-expression of G9a in vivo. As the effects on self-renewal cannot be experimentally verified, the authors should modify their claims in the manuscript.

*Reviewer #3 (Recommendations for the authors):*

1. The authors have addressed some of my previous comments. However, I find their responses to comments 12 and 13 vague and inadequate

2. Provide one or more citations for the claim that, "…multiple lines of evidence support the expression of LGR5 in the epithelial cell compartment." or, if that is not possible, then omit reference to LGR5 expression.

---

## [Author Response]

The reviewers have discussed the reviews with one another and the Reviewing Editor has drafted this decision to help you prepare a revised submission.As the editors have judged that your manuscript is of interest, but as described below that additional experiments are required before it is published, we would like to draw your attention to changes in our revision policy that we have made in response to COVID-19 (https://elifesciences.org/articles/57162). First, because many researchers have temporarily lost access to the labs, we will give authors as much time as they need to submit revised manuscripts. We are also offering, if you choose, to post the manuscript to bioRxiv (if it is not already there) along with this decision letter and a formal designation that the manuscript is "in revision at eLife". Please let us know if you would like to pursue this option. (If your work is more suitable for medRxiv, you will need to post the preprint yourself, as the mechanisms for us to do so are still in development.)This is generally a well-designed and carefully conducted study that is likely to be of interest to many both inside and outside of the field of lung tumorigenesis and normal lung development and tissue homeostasis. However, there are substantial conceptual issues and issues of data interpretation that the reviewers have raised regarding the research described in this manuscript. Although it may not be feasible for the authors to address every critique listed below experimentally, the authors should attempt to address the reviewers' critiques. Some of the key issues that were raised include:1. The results on tumor inhibition in vivo and proliferation of tumorspheres in vitro are not entirely surprising, given previously published literature by Zhang et al. (Molecular Cancer 17: 153 (2018)) showing that UNC0642 and/or shEHMT2 inhibit multiple pathways, including WNT signaling in cancer cell lines in vitro and in xenografts. The inhibitory effect on WNT signaling was ascribed to effects on the expression of APC2 resulting in increased destruction of β-catenin, a paper that authors of this manuscript do not reference. Rather, Pribluda et al. propose a model in which inhibition of G9a/EHMT2 activity in AT2 "stem cells" and in "tumor propagating cells" leads to increased WNT activity in a subset of AT2 cells and tumor cells as measured by Axin2 expression. In the case of AT2 cells this WNT pathway activation is proposed to drive the terminal differentiation of cells into a fully mature, non-proliferative AT2 cell. The authors argue that this loss of self-renewal and the potential to differentiate into AT1 cells limits tumorigenesis driven by the combination of mutationally-activated KRAS(G12D) and silencing of TP53.

To our knowledge, there are only two reports of G9a-mediated WNT regulation – one was retracted and therefore we will not comment on it (Cha et al., NCB 2016; PMID: 28008183). The Zhang et al. work (Zhang et al., Mol Cancer 2018; PMID: 30348169) which the reviewer refers to in his comments, relied on three cell tumor cell lines, A549, H1299 and H1975, to determine the role of G9a in this disease context. Indeed, the authors report that G9a inhibition impairs cell growth and they attribute this effect to Wnt inhibition. In our studies, we do not observe Wnt inhibition upon G9a manipulation in either the tumor-propagating (TPC) or non-TPC populations (Figure 4A). We view this report as a unique single context, represented by effects seen in three lung cell lines that notably all harbor mutations in ARID4A, a protein responsible for HDAC recruitment.The range of response to G9a perturbation varies widely across lung cell lines (https://depmap.org/portal/gene/EHMT2?tab=overview) and therefore it is unknown how the effects in these three cell lines apply. We do not believe that this report retracts from our work. We want to reiterate that we are studying a role for G9a in a distinct cancer context – one not observed in cell lines. Autochthonous primary tumors, known to harbor a unique and rare tumor propagating population (TPC) responsible for long-term regenerative capability. This biological situation is not represented in established cell lines. Furthermore, we also extend these findings beyond the tumor context and into the normal lung. The fidelity between normal lung and the tumor context strengthens our work and provides a broader applicability for G9a function outside of lung cancer. We agree with the reviewer that this work should be cited and discussed and have added the Zhang et al. reference and the above discussion to our manuscript.

More recently, an additional report published in *eLife* outlined a link between G9a and Wnt pathway suppression, similar to our own work (Ananya et al., EHMT2 epigenetically suppresses Wnt signaling and is a potential target in embryonal rhabdomyosarcoma. *eLife* 2021). Ananya et al. described a role for G9a*-*mediated suppression of Wnt signaling in Rhabdomyosarcoma through activation of DKK1, representing a distinct mechanism of G9a/Wnt regulation from what our work describes. DKK1 is not expressed in AT2 cells (Arun et al. Sci. Advanced Medicine 2020), however, the work still reinforces the concept of a role for G9a in suppressing Wnt signaling and we therefore also reference and discuss this work.

Lastly, regarding the role of p53 in our work, while indeed the tumor biology work is performed in the absence of p53 (KP GEMM), the Wnt modulation we observe in primary AT2 cells harbors intact p53. Therefore, the Wnt activation observed upon G9a loss appears to be independent of p53 status.

2. Others have shown that either pharmacologic or genetic inhibition of WNT signaling has striking inhibitory effects on lung tumorigenesis driven by the RAS/BRAF pathway. In that context, the authors should consider the results presented in Juan et al: "Diminished WNT->β-catenin->c-MYC signaling is a barrier for malignant progression of BRAFV600E-induced lung tumors" (Genes Dev. 2014 28: 561-75). In addition, Pacheco-Pineda et al., "Wnt/B-catenin signaling accelerates mouse lung tumorigenesis by imposing an embryonic distal progenitor phenotype on lung epithelium" (J Clin Invest 121: 1935-1945) further supports the hypothesis that elevated WNT signaling promotes lung tumorigenesis. This research clearly indicates that an insufficiency of WNT signaling is inhibitory to lung tumorigenesis, which stands in contrast to the authors contention that sustained WNT signaling promotes differentiation of TPC cells to a more differentiated AT2 phenotype thereby suppressing lung tumorigenesis.

We agree with the reviewer's comments and we would like to explain what seems to the reviewer as an inconsistency of our data with the literature. We are not disputing a requirement for Wnt signaling in maintaining stemness in the lung or in tumor propagation. While the reviewer is correct that Wnt has already been implicated as being required in these processes, in all those cases WNT signaling is not “modulated” per se, but either completely inhibited or constitutively activated.

In our study, we report an increase in basal Wnt signaling/Axin2 expression in response to G9a inhibition from the baseline steady-state level. We can directly link the outcome of this Wnt signaling increase to a cell differentiation effect through mechanistic studies showing a role for G9a in repressing B-catenin on TCF4 promoters of those same differentiation genes. Importantly, the impact of G9a functional impairment in our studies – either through knockdown, enzymatic inhibition or genetic deletion is consistent – increased Wnt signaling in BOTH TPC and in alveospheres leading to a cell state phenotype, indicative of a role for Wnt signaling in regulating AT2 biology.

These findings are not contrary to our reported data supporting a requirement for Wnt signaling – it is distinct, in that we are describing a requirement for a specific level of Wnt activity to maintain stem-like properties within those cells. In our setting, G9a is functioning to selectively regulate Wnt signaling to maintain a distinct threshold – that when lost, Wnt signaling is elevated above the set point leading to differentiation and a change in cell fate and capabilities.

We refer the reviewer back to our discussion where we addressed the differences between our findings and the previously reported ones, concerning the effects of pan inhibition/activation of Wnt signaling on tumor growth. We now include *Juan et al.* in with the previously reported findings and hopefully clarify the distinction between our work and that of others.

Please see below for the amended discussion:

“Of note, genetic cooperativity between Ras and Wnt signaling pathways has been reported (Juan et al., 2014; Pacheco-Pinedo et al., 2011) and linked to cell fate effects. In the context of Kras-mutant lung tumor initiation, genetic activation of Wnt signaling using mutant β catenin within Scgb1A1+ cells leads to a distal cell fate change consistent with our results. However, in contrast to our AT2-derived tumor-initiation model, results in enhanced tumor formation within Scgb1A1+ expressing cells (Pacheco-Pinedo et al., 2011). Notably, this work differs in the cell of origin, and in that regard, it is not yet clear how distinct, alternate thresholds of Wnt signaling contribute to determining cell fate within different cell lineages in the lung. Moreover, Pacheco-Pinedo et al., used a stable form of β catenin, whereas loss of G9a activity leads to signaling of chromatin-bound β catenin, which likely differ in both signal strength and duration. Others have reported the necessity of Wnt signaling to maintain tumors(Juan et al., 2014) and suggested that Wnt gradients might be critical for self-renewal, in line with the necessity for a specific Wnt threshold to support this process (Tammela et al., 2017).”

In conclusion, our data is not inconsistent with published work, nor does it dispute those findings. Our study is demonstrating that G9a is functioning to maintain a distinct basal threshold of Wnt.

Major comments:1. Throughout the manuscript, the authors manipulate their genes/pathways of interest largely in one direction, by inhibiting/depleting G9a, or by stimulating WNT signaling, without looking at the effects of over-expressing or rescuing G9a, or inhibiting WNT signaling. Their findings would be considerably strengthened if they could show that overexpressing G9a and inhibiting WNT signaling produced the reverse outcomes.

We show consistent effects of G9a on AT2 biology by three different approaches;(1) two shRNA constructs, (2) a pharmacological approach and (3) genetic deletion of G9a in vivo. By relying on three orthogonal approaches, the results support our conclusion that the effects on G9a biology are acting through modulation of G9a.

We did consider how to experimentally inhibit Wnt inhibition, as suggested by the reviewer, however, we show that G9a control on Wnt signaling occur directly on chromatin-bound β-catenin and therefore lies downstream of both Wnt-receptor ligand interaction and the β-catenin destruction complex, so inhibition of Wnt signaling upstream of these signaling events would not rescue G9a-mediated effects on chromatin. In addition, our work implicates an induction of Wnt above basal Wnt signaling. Experimentally, we are unable to impair only the G9a-induced suprabasal Wnt activity whilst maintaining the required basal Wnt signaling level.

As outlined above, the requirement of Wnt signaling has been previously explored for lung homeostasis and in the context of KRAS tumorigenesis, however these experiments were designed to explore complete inhibition of Wnt signaling. Indisputably, complete lack of Wnt signaling impacts all cell types, yet these studies do not address the requirements or modulation of thresholds of Wnt signaling between discrete populations. Therefore, due to the requirements of a basal Wnt signaling level for both TPCs and non-TPCs, as well as in AT2 stem cells and differentiated AT2 cells, Wnt inhibition experiments were not feasible.

As an alternative orthogonal approach to G9a inhibition, we relied on titrated activation of Wnt signaling (Figure 4 C-F) to solidify the link between G9a-mediated WNT signaling and cell fate outcomes. In these experiments, pharmacological activation of Wnt signaling using two doses of the GSK3β inhibitor CHIR99021, demonstrated impaired TPC stemness and reduced tumorsphere self-renewal, consistent with our data using G9a inhibition. Moreover, corresponding Wnt pathway activation and transcriptional activation of AT2 marker genes was observed at secondary passage (Figure 4E), with a concordant dose-dependent increase in SPC surface expression (Figure 4F). The consistency between direct WNT activation and cell fate outcomes, together with Wnt activation through the myriad of G9a perturbation approaches (ie. pharmacological, short hairpin and genetic deletion) further solidifies our hypothesis.

2. In the authors' tumorsphere assays, they have not ruled out the effects of G9a inhibition on cell proliferation and/or apoptosis, or assessed the proportion of CD24+ ITGB4+ NOTCH(hi) cells by FACs. These experiments are absent for both the in vitro and transplant experiments. These findings contradict previous literature that suggests that G9a is lost in a different TPC population and G9a inhibition increases another TPC marker, SCA-1 (Rowbotham et al. 2018). The authors should explain if these populations/markers are affected in their experiments and address any differences that they have observed.

We thank the reviewer for this question and would like to draw the reviewer’s attention to several figures to help address this query. We report decreased proliferation and reduced apoptosis in tumorspheres following G9a inhibition (Supplement figure 3-1) and demonstrate the proportion of CD24+ ITGB4+ NOTCH(hi) cells by FACS (Supplement figure 3-5). We further show the molecular (Figure 3A) and morphometric features (scanning electron microscopy in Figure 3D-E) that support AT2 differentiation as the final outcome. Additionally, we show an increase in cell fate differentiation markers in tRFP-sorted tumor cells derived from primary transplants. (Supplement figure 3-4). Figures are included below the next paragraph for the ease of the reviewer.

The reviewer also raises the important distinction between the “TPC” populations identified by Zheng and colleagues (original TPC finding, of which our work builds upon) versus Rowbotham and colleagues. In Rowbotham et al. the CD24+ Sca-1 population consists of >15% of all epithelial cells in KP tumors. However, the population discovered by Zheng et al., CD24+/ITGB4+/Notch^HI^ constitutes ~1-2% of the epithelial population. The difference between these populations and their relative abundance makes it difficult to compare them. However, we show in our work that G9a is enriched in CD24+/ITGB4+/Notch^HI^ TPCs by two methodologies, RNA expression and protein expression. Notably, Rowbotham indeed showed that G9a transcript is 2-fold lower in Sca1+ cells, but no confirmation of correlative protein expression was demonstrated. Regardless of the definition of the sub-population of TPC being studied, G9a protein expression levels are more indicative to extrapolate a putative function of G9a biology in the subpopulation.

3. It is surprising that the increase in CD74 (Difference in SP-C is not significant?) is only restricted to the TPCs and is only ~2 fold. In Figure 3, Supplement 5 the data indicates that TPCs make up ~0.12% of the sphere culture. If this was the case the increase in SPC in the whole sphere culture would be insignificant. However, in Figure 3A SPC levels are significantly higher in the G9a inhibited samples and appear widely distributed across the tumorsphere suggesting that many cells and not just the 0.12-0.02% that are TPCs have increased SPC. How do the authors account for this observation?

We agree with the reviewer’s comments, however we don’t see it as inconsistent. G9a inhibitor is maintained in culture for 5-6 days, in which AT2-like TPCs give rise to progenitors that form these tumorspheres. The progenitors would continue to show sustained AT2 marker expression. Ex vivo the number of TPC is higher (0.5.-2%), which would further explain the elevated SPC expression.

4. It is perhaps surprising that the authors did find many changes in chromatin accessibility in their inhibited tumorspheres, whence they conclude that G9a inhibition does not alter chromatin in their cultures. In Figure 1E however they clearly show that levels of the chromatin modifications H3K9me2 and H3K9me3 are significantly reduced suggesting that the chromatin has been altered. The authors should account for this and rule out technical problems with their ATAC-Seq data. Indeed, although the data on β-catenin⌠RUVBL2 is interesting, it remains unclear whether this interaction is indeed brokered by changes in RUVBL2 lysine methylation.

Please note that chromatin accessibility measurements were assessed in TPCs and not in tumorspheres. Our aim was to assess G9a-specific changes within the TPC population itself to understand the mechanism by which G9a inhibition resulted in reduced regenerative capacity for this population – it is within this population that we did not observe significant chromatin accessibility changes upon G9a inhibition. We agree with the reviewer that one may expect more differences in chromatin if comparing a stem-like population to a progeny population, which is why we chose to focus on the “stem-like” population for this analysis.

However, to further clarify our results, even within the TPC population, we do find changes in chromatin accessibility in TPCs, as described in the manuscript: “accessibility increases at 331 sites and decreases at 57 sites upon G9a inhibition”. Although genes with impacted promoter accessibility could not be linked to the phenotype induced by G9a inhibition. We agree that it can be clarified better in the figure itself. As to Fig1E, we indeed show reduction of H3K9me2 and H3K9me3 marks, but this reduction is in tumorspheres – a more diverse population, including both TPCs and non-TPC progeny.

To assuage any further doubt of the ATAC-seq data analysis, we have included more detailed information methods regarding how the QC was performed:

“Assay for Transposase-Accessible Chromatin using Sequencing (ATAC-Seq) Cells were sequenced by Epinomics, and processed as previously described (Buenrostro et al., 2013). Paired-end reads were aligned to mouse reference genome GRCm38 using GSNAP (Wu and Nacu, 2010) Duplicate reads and reads containing substantial sequence homology to the mitochondrial chromosome or to the ENCODE consortium blacklisted regions were omitted from downstream analyses. Remaining reads were quantified according to the ENCODE pipeline with minor modifications: Accessible genomic locations were identified using MACS 2.1.0(Zhang et al., 2008) as insertion-centered pseudo-fragments (73bp with a width of 250bp). Peak significance (cutoff of p=1e-7) was determined on a condition-level pooled sample containing all pseudo-fragments observed in all replicates within each condition. Peaks independently identified as significant (cutoff p=1e-5) in two or more of the biological replicates were retained, using the union of all condition-level reproducible peaks to form the Encode atlas. The atlas consisted of 184,032 peaks with a median width of 266 bp (ranging from 250 bp to 1531 bp). 20.4% of peaks were located in promoter regions, 32.6% in intergenic regions, and 43.3% in introns. Chromatin accessibility within each peak per replicate was quantified as the number of pseudo-fragments overlapping a peak and normalized using the TMM method (Robinson and Oshlack, 2010). Differentially accessible peaks between control and G9a-inhibited samples were identified using a linear model, accounting for TPC and non-TPC and implemented with edgeR. Significance within a peak between G9ai-treated and control-treated samples was set to log2-fold change > 1.5 and FDR < 0.05. The fold-change was used as input for gene set enrichment analysis using the Hallmark MSigDB gene set collection, v6.1 (Liberzon et al., 2015). Integrative Genomics Viewer (IGV) was used for visualization of ATAC-seq peaks near genes of interest (Robinson et al., 2011).”

5. In the distal lung, WNT responsive Axin2+ AT2 cells have been demonstrated to act as progenitors which can regenerate a substantial fraction of the alveolar epithelium (Frank et al. 2016, Zacharias et al. 2018, Nabhan et al. 2018). Zacharias et al. suggest that SP-C+/Axin2+ cells are an alveolar progenitor distinct from AT2 cells and Nabhan et al. suggest that Axin2+ AT2 cells have stem cell activity. The authors are suggesting that in the context of a KRAS(G12D)-driven lung adenocarcinoma, WNT signaling and Axin2 expression have the opposite effect and oppose the stem cell activity of the TPCs. As this is in direct contrast to a study using the same mouse model which shows that WNT signaling supports the self-renewal and tumor propagation of TPCs, and inhibiting WNT signaling from the niche impairs tumor growth and progression (Tammela et al. 2017), the authors should test their mechanism in vivo or explain the discrepancy.

We thank the reviewer for this comment. As a reminder and as discussed similarly in comment 2 above, our work is not challenging the existence of these cell types, nor their requirement for Wnt signaling. These elegant studies all point to an absolute requirement for constitutive basal Wnt signaling. Our work is focused on describing how increases from this basal level can impact cell biology and fate. In fact, these references support the precise concept that differential Wnt signaling levels (ie. AXIN2 levels) distinguish cell types and their capabilities in the lung.

We observe that a specific level of Wnt activity is required to maintain stem-like properties within TPC and AT2 cells – performed by multiple means of G9a manipulation both in vivo (tumor context and normal lung tissue) as well as ex vivo (with primary KP tumor organoids and isolated AT2 cells). In our settings, G9a is functioning to selectively regulate Wnt signaling to maintain a distinct threshold – when that is lost, Wnt signaling is elevated above a set point leading to differentiation and a change in cell capabilities.

It is, indeed, an excellent question as to whether G9a manipulation would, for example, alter this rare Axin2-GFP+ AT2 cell population as described in Nabhan et al. In fact, Nabhan and colleagues show that increased Wnt signaling within their Axin2-GFP+ AT2 population prevents emergence of AT1 from AT2, which is consistent with our data in Figure 6 with Wnt signaling induction mediated through G9a loss.

6. The authors show the RUVBL2 and β-Catenin interaction is altered by G9a inhibition in a human cell line. Can the authors also show this interaction in their tumorsphere cultures or at least with a mouse KRAS(G12D)/TP53(Null) cell line for relevance to the rest of their results?

Further, we considered performing these mechanistic experiments in cell lines derived from KP GEMM tumors; however, pharmacological inhibition of G9a in these cell lines did not impact sphere formation, suggesting that these lines do not harbor the relevant TPC population to progeny relationship found in de novo KP tumors, precluding us from using cell lines for this purpose. See Author response image 1, showing no significant effect with G9a inhibitor in 3 KP derived cell lines.

**Author response image 1. sa2fig1:** 

We attempted immunoprecipitation from tumorspheres as the reviewer suggests; however, due to the fact that we are isolating TPCs from primary KP tumors, we were unsuccessful in obtaining sufficient material to perform this experiment fully. Therefore, we needed to rely on proximity ligation assay to allow us to explore the interaction in the TPC population.

7. In Figure 6 the authors should use a pharmacodynamic biomarker to show that the G9a inhibitor reaches the epithelial cells of the lungs and is actively inhibiting di- or tri-methylation of H3K9. The results are not consistent with the conclusion that AT2 stemness has been altered by G9a inhibition, as this has not been directed tested. The data from Figure 6B showing no difference in sphere forming ability suggests against this interpretation. The authors should passage their G9a inhibited alveolar spheres and test this directly. As the proportion of AT2 cells between the two cultures does not appear to be altered we might not expect to see a significant difference. The authors should perform statistical tests to show that the differences between cells with AT2 and AT1 markers are significantly different, even though they appear large. The results seem to suggest not that AT2 stemness is lost, but that the differentiation to AT1 cells is impaired, leading to a majority intermediate type cell (SP-C+/PDPN+) instead of a majority of AT1 cells. This would be consistent with existing literature showing that activating WNT signaling with the CHIR GSK3-β inhibitor increases the proportion of AT2 cells in culture whilst inhibiting WNT signaling increases the proportion of AT1 cells (Zacharias et al. 2018).

UNC0642 is more than 20,000-fold selective for G9a and GLP over 13 other methyltransferases (IC50 > 50,000 nM), more than 2,000-fold selective over PRC2-EZH2 (IC50 > 5,000 nM), and displays low cell toxicity (EC50 > 3,000 nM) (Liu et al., J Med Chem 2013). To address the reviewer’s question regarding in vivo exposure of compound, we have now included the pharmacokinetic data corresponding to the G9a inhibitor used throughout our studies and confirmed the in vivo exposures and metabolic stability of UNC0642 (Liu et al., J Med Chem 2013; Author response image 2). Following a single IP dose of 5mg/kg of the G9a inhibitor, the serum concentration remains stable out to 24 hours, consistent with the reported PK properties of this molecule. Unfortunately, we do not have in vivo pharmacodynamic (PD) data beyond the consistent AT2 markers described. presently.

**Author response image 2. sa2fig2:** Pharmacokinetic behavior of UNC0642: (A) concentration from plasma measured at 8 time points with 3 animals per time point after a single 5 mg/kg intraperitoneal dose of UNC0642 using a PEG400/H2O (60/40) formulation. (B) Summary table of observed parameters from Liu et al., 2013 and Pribluda et al. Note no formulation information is available from Liu et al., 2013.

In addition, we refer the reviewer to supplement 3-4, where we show an increase in AT2 markers in vivo as a result of G9a inhibition*.* This observed increase in AT2 markers is consistent with our results reported in the alveosphere and tumorsphere cultures, where in the latter, we assessed H3K9 di and tri methylation as a PD readout for G9a inhibition. Importantly, we consistently observe increases in Axin2 and AT2 markers, using genetic deletion and shRNA approaches as a means to inactivate G9a both ex vivo and in vivo (shRNA, pharmacological inhibition and genetic deletion). Therefore, we are strongly convinced that G9a is inhibited in the lung with treatment**.**

As the reviewer mentions, we do not see differences in sphere forming ability with the AT2 alveosphere cultures, but we disagree that this means our interpretation is flawed. The plasticity of AT2 cells is measured in part by their ability to differentiate to AT1 cells, and indeed despite the lack of change in sphere forming ability, we do observe a significant increase in both Wnt target genes and AT2 cell-lineage markers, consistent with our early observations in the tumor context. Furthermore, we confirm the increase in SPC+/PDPN+ double-positive cells within alveospheres, using a genetic deletion of G9a in vivo. Lastly, we want to draw the reviewer’s attention to *Zacharias et al. Nature 2018*, where no change in sphere number is detected after Wnt signaling activation (see AT2 alveospheres treated with CHIR vs. DMSO, extended data Figure 9 in that paper). Therefore, given these consistencies, we do not think our interpretation contradicts our results. Moreover, the assay, as performed, is the standard practice used by others in the field to determine stemness of AT2 cells (Nabhan et al. Science 2018; Zacharias et al. Nature 2018).

We agree with the reviewer’s comment that reduced stemness and impaired differentiation could both generate the outcome observed. Because AT2 stem progenitors are believed to be the population responsible for forming alveospheres, we interpreted impaired differentiation as indicative of reduced stemness. We now include language to cover both possibilities by referring to reduced “plasticity”.

Text changes are as follows:

“Together these data indicate that G9a inhibition impairs the plasticity within the AT2 progenitor pool by disrupting proper AT1 cell fate transition.”

And “Together these data indicate that G9a inhibition impairs the plasticity and complete differentiation within the AT2 progenitor pool by disrupting proper AT1 cell fate transition.”

8. The authors administer Cre adenovirus to a G9a(fl/fl) reporter mouse to further study the effects of G9a loss in vivo. However, the authors do not describe this allele in their Materials and Methods nor do they cite an original publication of such an allele/mouse. (Perhaps this is the relevant citation: Lehnertz et al., 2010 Activating and inhibitory functions for the histone lysine methyltransferase G9a in T helper cell differentiation and function. J Exp Med. 207: 915-922). However, if this mouse is not published the authors should provide additional information on the provenance of this allele/mouse. The authors should demonstrate that G9a expression is successfully silenced in this tumor model, or if the mechanism they propose is active in vivo, whereby G9a deletion in tumors would be presumed to establish continuous WNT signaling to maintain AT2 differentiation and either prevent tumor initiation or retard tumor growth.

We apologize for not providing the targeting details of these new alleles. This is now included in the supplemental information together with a description.

**Author response image 3. sa2fig3:** 

*Ehmt2* expression vector was constructed by introducing a Frt-PGK-em7-NEO-Frt as a selection marker. loxP sites were introduced into the 5’ and 3’ homology sequences flanking the targeted exons (genomic location 34908772-34912090 and 34916244-34918676 respectively). The loxP sites, flanked exon 25-27*,* constituting the SET catalytic domain of *Ehmt2*. 5’ and 3’ extra genomic regions were used to design PCR primers to validate the targeted deletion which generates a 2.2kb fragment upon Cre recombinase administration. Conditional gene deletion in the adult was generated in the lung upon Adeno-Cre administration.For supportive data that G9a is lost following Cre exposure, please see supplemental figure 6-3.

9. The statistical analysis of the SPC+/PDPD+ cells following hyperoxic injury should be reassessed. The >10,000 cells the authors enumerate are not independent of each other and should not be analyzed by a t-test in this manner. An n>10,000 is too high for a t-test and will almost always result in p<0.05, rendering the statistical analysis meaningless. A more relevant way to analyze this data is to average the % SPC+/PDPN+ cells per mouse and then perform a statistical test on these values (n=6), assuming each mouse was independent of the others. A t-test would be appropriate if the data are normally distributed, and if not, a non-parametric ranked test such as Mann-Whitney.

We repeated the analysis, averaging the data per mouse as suggested by the reviewer. The change remains significant using t-test for statistical analysis. We now use this figure and analysis in the manuscript.

10. If the mechanism suggested by the authors is correct, then they should show that G9a is deleted in the alveolar cells that become SPC+/PDPD+ after hyperoxic injury and not in the cells that escape G9a deletion.

We are not sure what the reviewer is referring to by “escape” in a genetic deletion in the setting of the hypoxic normal lung experiments. G9a deleted cells are labeled with tdTomato reporter (Supplemental figure 6-4), where we have verified that indeed G9a is deleted in these cells (Supplemental figure 6-3). The analysis was conducted only in tdTomato-positive cells and the legend and graph labels now reflect this for further clarity.

Whilst the authors do present evidence that alveolar cell fate appears to be affected by G9a, the 'stemness' of the AT2 cells has not been measured either directly (by lineage tracing their progeny and quantifying their contribution to the new epithelia) or indirectly by studying the dynamics (i.e. timing) of regeneration following injury in G9a(fl/fl) vs normal mice.

The reviewer raises a relevant point regarding the use of “stemness” to describe the observed cell fate changes in response to G9 perturbation. Furthermore, we do not have lineage-tracing evidence in our in vivo AT2 hypoxia experiments to definitively support the AT2-to-AT1 relationship. Moreover, we agree that both the lineage tracing and the dynamic experiments are important follow-up experiments. To ensure that our conclusions are balanced, we removed mention of “stemness” and instead use “plasticity” to more accurately reflect the data. Additionally, we report only on the increased presence of the double positive cells as supportive of our ex vivo data and we incorporate the caveats described above.

“Studies evaluating the impact of G9a loss at later stages of the tissue repair process will further contribute to our understanding of its role in epithelial cell fate decisions in the lung.”

11. Throughout the manuscript the authors refer to AT2 "cell fate identity" and to the "trans-differentiation" of AT2 to AT1 cells. This is quite confusing since there has been important work on the role of transcription factors such as SOX2 and NKX2.1 on determining the "identity" of epithelial cells in NSCLCs (e.g. Tata et al. 2018 Dev Cell 44: 679-693 and Snyder et al. 2013 Mol. Cell. 50: 185-99). In this previous research a change in identify involves true switching of fate from lung to non-lung (e.g. intestinal or esophageal) cell types. Consequently, it is unclear whether the differentiation of AT2 cells to AT1 cells should be considered "trans-differentiation" or loss of "cell fate identify" but rather the normal behavior of AT2 cells in homeostasis and repair of the adult lung.

We agree with the reviewer that this terminology is important and should be used consistently throughout. We have modified the text to refer to AT2-AT1 fate as differentiation. We have removed all transdifferentiation language throughout the manuscript.

See below for example:

“Deletion of b-catenin in AT2 cells leads to AT1 cell fate transdifferentiation”.

Changed to

“Deletion of b-catenin in AT2 cells leads to AT1 cell fate differentiation”.

12. The authors should consider the confusing fact that two papers that describe active WNT signaling in a subset of AT2 "stem cells" in the normal mouse lung (Nabhan et al. 2018 and Zacharias et al. 2018) report very different proportions of these cells (~3% vs. ~30%). Indeed, the authors assume that there is a fixed subset of AT2 stem-like cells and that these are maintained in a "plastic" self-renewing state by the correct level of WNT signaling. Perhaps AT2 cells can oscillate between states in which they are more or less differentiated, and that it takes longer for the fully differentiated cells packed with lamellar bodies and associated secretory machinery to start to divide and/or give rise to AT1 cells? Indeed, there is little evidence in vivo that KRAS-driven tumors can only arise in Axin2+ AT2 "stem cells", albeit that WNT signaling is essential for both KRAS and BRAF-driven lung tumorigenesis.

We agree with the reviewer’s comments. It has been clearly shown that Wnt is essential for tumorigenesis. We do not dispute that nor are we claiming the “AT2 stem cells” are the cell of origin of KRAS/BRAF driven adenocarcinomas.

13. To support their model the current authors should consider testing whether their tumorspheres contain both AT2 and AT1 cells and whether treatment with UNC0642 specifically reduces differentiation of AT1 cells and does not affect other pathways regulating cell proliferation and survival, as proposed by Zhang et al.

The tumorspheres are initiated through isolation of tumor cells, excluding AT2 and AT1 cells; however, these spheres stochastically express AT2 and AT1 markers, nevertheless we assessed the changes using an AT1 transcriptional signature and immunofluorescence and we found no significant changes in AT1 markers after G9a inhibition (See Figure 3). The effect of G9a on proliferation and survival had already been answered in previous comments (Please refer to our answer from comment no.2).

The experiments in which Pribluda et al. test the effect of inhibiting EMHT2 on primary mouse AT2 cells in organoids (alveolosphere) culture are confusing. The authors state, "During alveosphere formation and expansion, emerging cells express transcriptional and surface markers consistent with an AT2 to AT1 cell fate transition". A "cell fate transition" was not ascribed to the normal differentiation of AT2 to AT1 cells by the original authors.

Agree. The statement has been changed to be consistent with the biology reflected in the assay and now reads: "During alveosphere formation and expansion, emerging cells express transcriptional and surface markers consistent with an AT2 to AT1 cell differentiation".

The fact that the treated spheres are smaller than normal may reflect reduced cell proliferation – however, this was not assayed and the morphology of the spheres and immunohistochemistry is not shown.

Indeed, reduction in overall proliferation could explain these results if AT1 cells were exclusively dying. However, we show that the fraction of AT2 cells remains similar in both G9a-treated cultures and controls, whereas it was previously shown that once AT1 cells are generated they exit the cell cycle (Barkauskas et al. JCI 2013, figure S6). Therefore, we interpreted that the increase in intermediate double positive cells comes at the expense of proper AT1 differentiation.

The authors state "Interestingly, while the proportion of SPC+/PDPN- AT2 cells (SPC+/PDPN-) remained unchanged, we observed a marked increase in double positive cells (SPC+/PDPN+) from 15% to 62%. Together these data indicate that G9a is required for differentiation of the AT2 progenitor pool and for complete and proper AT1 cell fate transition". However, the significance of dual positive cells in vivo or in vitro is not known and it is unclear how the lack of change in the proportion of AT2 cells can be interpreted as reduced differentiation of AT2 cells.

The reviewer is correct that we do not fully understand the significance of the dual positive population. However, the sentence proceeding the referenced excerpt stated “By analyzing expression of surface markers indicative of these fates (Desai et al., 2014), we observed that G9a inhibition significantly reduced the proportion of AT1 cells (SPC-PDPN+) from 67.2% to 24.4% (Figures 6C—figure supplement 2)”. We understand that in this assay the reduced AT1 production is indicative of impaired differentiation, consistent with other reports using this assay system (Nabhan et al. Science 2018; Zacharias et al. Nature 2018). We conclude that the differentiation from AT2 to AT1 is impaired and provide further details regarding the double positive cells. While the double population clearly exists in the alveolar assays, it is typically not reported in these types of assays – rather the focus is primarily on AT1 and AT2 cell fates.

Finally, it should be noted that neither LGR5 nor LGR6 are not normally expressed in lung alveolar epithelial cells in vivo – rather expression is confined to mesenchymal cells.

The reviewer is referring to Figure 6D, where expression of Axin2, Lgr4 and Lgr5 is used to confirm Wnt pathway activation in primary AT2-derived murine spheres following G9a inhibition. We presume that the reviewer is basing his statements of lack of Lgr5 on the data published by Lee et al. Cell 2017, where Lgr5 was not detected in the epithelial population by single cell RNA-seq. Because of dropouts, genes undetected in scRNA-seq assays cannot be considered as a confirmation of lack of expression, especially for lowly expressed genes. Indeed, expression of Lgr5 in our own bulk RNA-seq dataset was very low (2-17 reads out of 30-40M reads per sample), which likely means that Lgr5 cannot be detected in standard scRNA-seq assays. Although Lgr5 is a low-expressed gene, it is readily observable by qPCR, most likely due to the amplification steps included in this procedure. While Axin2 and Lgr4 increases suffice to demonstrate wnt pathway activation, we would like to retain Lgr5 as multiple lines of evidence support its expression in the epithelial cell compartment.

[Editors' note: further revisions were suggested prior to acceptance, as described below.]

The manuscript has been improved but there are some remaining issues that need to be addressed, as outlined below. Indeed, as you will see from the reviews below, two of the three reviewers are willing to accept the manuscript, whereas the third reviewer has raised a number of additional concerns, not the least of which is the fact that only one of the shEHMT2 hairpins reduces sphere formation despite the fact both hairpins elicit strong reduction of EHMT2 expression. Hence, all three reviewers are concerned about the possibility of off-target effects. Hence, in order to satisfy all three reviewers, and to allow us to accept your manuscript, we ask that:1. You address the comments raised by Reviewer #2 on a point-by-point basis.

We have provided a point-by-point response to R2.

We incorporated three text changes to satisfy concerns of R2. Changes are outlined in the point-by-point response below in Point 2 and in Point 14.

2. That you more explicitly state in the text a concern regarding the issue of the shEHMT2 hairpins.

We have added text to more clearly point out that while both hairpins reduce sphere formation, only one reaches statistical significance.

Modified language:

“Both UNC0642 treatment or short hairpin RNA (shRNA)-mediated depletion of Ehmt2 similarly impaired secondary sphere formation (Figure 1F-I), albeit only shEHMT2.2 reached statistical significance, establishing a requirement for EHMT2 activity in TPC self-renewal.”

3. That you address R3’s comments regarding citations that document expression of LGR5 in the epithelial compartment.

The reviewer misunderstood our response to his query regarding LGR5 expression that we detect in our studies. The “multiple lines of evidence” is in reference to our own data contained in this paper, RNAseq and q-PCR. Given the paucity of data on this topic and the fact that LGR5 is a well-established WNT target gene, we are compelled to retain this information for the field, if the editor agrees.

Reviewer #2 (Recommendations for the authors):Pribula et al. have revised their manuscript “EHMT2 methyltransferase governs cell identity in the lung and is required for KRASG12D tumor development and propagation. In a lengthy response to the first round of reviews, the authors make some changes to improve their manuscript, particularly with regards to the interpretation of their data. However, they do not appear to have attempted any of the experiments suggested by the reviewers to improve their manuscript, particularly with regard to testing the effects of G9a inhibition/knockdown on the Sca-1+ population of KP tumors in their systems, which has been published to interact differently with G9a loss than the authors report, and connecting the mechanism established in A549 cells to their phenotypes in tumors and alveospheres.Furthermore, in response to a minor point, the authors have highlighted a significant problem with their manuscript that was overlooked in the previous round of revisions. In Figure 1I it was noted that only one hairpin was denoted as significantly reducing tumorsphere formation. Initially assumed to be a mistake, the authors clarify that indeed only one hairpin significantly reduced tumorsphere formation. This leaves only one valid hairpin showing the same phenotype as G9a inhibition and as shRNA’s are notorious for off-target effects, it cannot be concluded from just one hairpin that effects were due to observed reduction in EHMT2 mRNA.This concern carries over into figure 2 where inducible hairpins are used to study the effects of EHMT2 knockdown on transplanted tumorigenesis in vivo. In the description of these experiments, it is not clear if the same two hairpins were used, and if both hairpins independently produced the same phenotype. Indeed, from the legend of Figure 2E, it appears that only one hairpin was used (shEHTM2 2.1). All shRNA hairpin experiments must be completed with at least two, and ideally 3+ independent hairpins to ensure that the phenotypes observed are not due to off-target effects. The sequence of the hairpins used does also not appear to be in the manuscript. Because these are serious concerns regarding experiments key to the manuscripts claim, we must conclude that the manuscript is still not suitable for publication in it’s current state.

In order to support our conclusions presented in our manuscript we utilized multiple experimental methodologies to target EHMT2. We show consistent effects of EHMT2 on AT2 biology by three different approaches; (1) two shRNA constructs, (2) a pharmacological approach and (3) genetic deletion of EHMT2 in vivo using a new genetic mouse model. By relying on three orthogonal approaches, the results support our conclusion that the effects on EHMT2 biology are acting through modulation of EHMT2.

We acknowledge that only one hairpin results in statistically significant reduction in sphere formation in Figure 1I (shEHMT2.2); however, the lack of statistical significance does not equate to lack of biological relevance. Both hairpins significantly reduce *EHMT2* transcript and both result in a clear reduction of sphere formation. To highlight this perceived discrepancy and lack of statistical power, we now include text in the manuscript regarding the lack of statistical significance for one hairpin in this experiment and include the actual p-value (p=0.09) in the legend.

Specific points:In response to point 1. the authors note that in discussing Zhang et al. 2018, and the results that UNC0642 and shEHMT2 inhibited wnt signaling this study relied on three human cancer cell lines, A549, H1299 and H1975. They argue that the results of this paper are context dependent on these cell lines, and not applicable to their primarily mouse GEMM studies. However, the biochemical experiments linking G9a to Wnt regulation in Pribula et al. are performed in A549 cells and not in primary GEMM tumors, organoids or AT2 cells (For obvious practical reasons). As the authors establish their G9a-wnt mechanism in this same cell line where g9a inhibition was shown to suppress wnt signaling, this should be addressed. If A549 cells are such a different biological context to their GEMM model (ARID4A mutations etc), then can they expect that their mechanism found in A549 cells carries through to GEMM tumors but the inverse finding on wnt signaling activation does not? Do the authors observe the same results with G9ai in A549 cells as Zhang et al.?

The reviewer raises a valid point regarding the differences seen in our work vs. Zhang et al. 2018, whilst both using the A549 cell line. In our work using A549, we observed that EHMT2 perturbation results in increased Axin2 expression, consistent with our hypothesis, please see below our results in A549, therefore we proceeded to work with A549 to help address mechanistic questions not feasible to address in other systems, as the reviewer points out.

**Author response image 4. sa2fig4:** 4 day knockdown with siRNA hairpin targeting G9a.

Notwithstanding, the A549 cell line was shown to harbor strains that are genetically and transcriptionally distinct from each other, which in turn could result in different drug responses (Ben-David et al. Nature 2018), thus making it difficult to compare between our conclusions with the A549 cells line vs. Zhang et al. Our claims related to distinct context were referring to this possibility.

2. The authors make a complex argument that G9a regulates a precise level of wnt activity, when G9a is inhibited an increase in wnt activity drives tumors cells to a differentiated AT2 cell state. This idea of a distinct Wnt threshold is intriguing. This concept and the context of pacheo-Pinedo et al. and Juan et al. should be addressed much earlier in the manuscript when wnt signaling is first introduced and not only in the discussion. The Wnt-threshold concept suggests that either activating or inhibiting wnt could both be deleterious or supportive to lung tumors, depending on context (i.e cell of origin) which would make clinical use of wnt inhibitors/activators a challenging prospect.

We agree with the reviewer’s comment.

We added this text in Results section:

“Additionally, genetic cooperativity between Ras and Wnt signaling pathways has been reported (Juan et al., 2014; Pacheco-Pinedo et al., 2011) and linked to cell fate effects. In this case, mutant β catenin within Scgb1A1+ cells caused a distal cell fate change and enhanced tumor formation within Scgb1A1+ expressing cells (Pacheco-Pinedo et al., 2011).”

1. The authors regret that they were unable to experimentally inhibit wnt signaling, arguing that because in their model, because G9a functions downstream of where most wnt inhibitors act, e.g. on ligand interactions and the B-catenin complex, and because they argue that only suprabasal wnt activity leads to the observed phenotype. If this is interpretation is correct, it would actually present an elegant method to test their model, as wnt inhibition (very easy to technically administer and can be carefully titred to the correct level) should therefore not be able to rescue the effects of G9a inhibition in their system. We believe that this experiment should be completed and reported as it is simple and would greatly strengthen their claims.

We thank the reviewer for raising this point again and we will try to further clarify our previous response. The mechanism we are exploring postulates that EHMT2 activity regulates basal steady-state chromatin-bound β catenin-mediated transcription by modulating β catenin-RUVBL2 interactions. Inhibition of upstream signaling with a Wnt inhibitor as the reviewer suggests would therefore not address the EHMT2 effect on the chromatin-bound β-catenin signaling pool. The reviewer is suggesting that we perform an experiment where we inhibit Wnt signaling by a means that would not include the pool of EHMT2/b-catenin of interest and demonstrate that has no impact our phenotype. Due to the requirements of a basal Wnt signaling we do not believe that this experiment is feasible and would not contribute further to our understanding of β catenin-mediated transcriptional activity upon EHMT2 inhibition.

2. The authors highlight some tumorsphere data showing that both proliferation and apoptosis are reduced. The purpose of asking for these data was to confirm if the effects of G9a inhibition were truly on stemness which leads to more organoids forming, or because G9a inhibition might reduce proliferation or increase apoptosis of the tumor cells. The results highlighted from figure S 3-1 show that whilst apoptosis is down, proliferation is also strongly reduced. Therefore are the results from G9ai on tumor growth, formation etc due to changes in proliferation rather than AT2 cell differentiation? Or do the authors consider that these are indistinguishable?

The reduced proliferation and cell death in tumorspheres led us to explore a differentiation as a possible outcome upon G9a inhibition. In tumorspheres, we show that differentiation leads to loss of self-renewal by two different sets of experiments: (1) using G9a inhibitors or shRNA and (2) by stabilizing β catenin using a GSK3b inhibitor. In both sets of experiments, we detect evidence of cell differentiation prior to loss of self-renewal. Therefore, we conclude that effects resulting in sphere size changes are a consequence of cell differentiation.

3. Whilst the discussion of the different TPC populations described is appreciated and adds important context, noting that they have different abundances. The question of whether the other TPC marker, Sca-1 and the Sca-1+ fraction is affected in their experiments remains unaddressed when a simple QPCR for Sca-1 in any of their tumor systems could help to address this.

To adequately address the abundance of the alternative TPC population of interest to the reviewer, would require running additional experiments using a newly optimized flow cytometry panel that included those surface markers. This alternative TPC population was not the focus of the work, moreover the outcome of those experiments would not impact the results or conclusions of the manuscript.

However, our RNAseq dataset will be publicly available for query for any and all genes of interest.

For convenience to R2, we include in Author response image 5 the results of *Sca-1* expression from EHMT2 inhibitor treated ex vivo tumor spheres. EHMT2 inhibition did not alter *Sca-1* expression.

**Author response image 5. sa2fig5:** 

4. We appreciate the reviewer's explanation, highlighting the difference between the in vivo and tumorsphere results. To incorporate this explanation into the text, the authors should mention that the broadly increased SPC expression observed in the cultures is likely from progenitors and differentiated tumor cells whilst in vivo only TPCs show an increase in genes associated with AT2 differentiation.

We thank the reviewer for the comment and it is now incorporated into the main text.

5. The authors helpfully remind the reviewer that chromatin accessibility was measured in TPCs and not tumorspheres, where they demonstrate that H3K9me2 and H3K9me3 are, as expected, reduced by G9a inhibition. Although they find very few changes in chromatin accessibility in TPCs, the method they describe is appropriate and if carried out correctly should find differences if they are there. This explanation seems to suggest that chromatin modifications would not be changed in TPCs and differences are restricted to tumorspheres "a more diverse population including both TPCs and non-TPC progeny". The authors would seem to be suggesting, therefore, that unlike tumorspheres in culture, H3K9me2 and H3K9me3 are not reduced by G9a inhibition in TPCs. This would be a surprising result, and at the very least should be investigated by the authors.

Indeed, the reviewer captured the results correctly. Our data indicated that the EHMT2 inhibition-induced phenotype is driven by a mechanism independent of changes in chromatin accessibility. Indeed, H3K9me2 and H3K9me3 marks are reduced by EHMT2 inhibition in TPCs, but few changes in chromatin accessibility were observed in the TPC subset. Moreover, pathway enrichment analysis on changes mediated by EHMT2 inhibition did not explain the observed phenotype. These results led us to propose an alternate hypothesis that, as the reviewer points out, is very interesting and was rigorously investigated and presented in our work where we show that Ruvbl2 acts as a non-histone substrate, regulated by EHMT2 activity and affects β catenin-mediated AT2 gene transcription.

6. In the explanation provided the authors refine their explanation of the effects of G9a and WNT signaling on TPCs, to align their conclusions with the existing literature. As mentioned in earlier rebuttal points, they interpret their data to suggest that G9a restriction of Wnt signaling is supportive of TPC function, and if G9a is lost or inhibited then excess wnt signaling can lead to loss of TPC function. This represents an interesting finding, although the authors perhaps overinterpret their findings when they suggest that G9a regulates Wnt to a distinct threshold (author's emphasis).With the authors data and the existing literature suggest that either too little, or now too much Wnt signaling can be deleterious to TPCs and AT2 cells. Whether there is a distinct threshold of wnt signaling for TPCs/AT2 cells to maintain proper function, and more importantly, what this threshold might be is interesting to speculate about, but goes beyond what the author's data show. It would take a considerable amount of experimentation to demonstrate that such a distinct threshold exists, more than could be reasonably expected in a revision of this manuscript. The authors should amend the description of their results and the discussion appropriately.

To clarify for R2, we do not state anywhere in our manuscript that EHMT2 regulates a distinct threshold of Wnt, but given the questions by the reviewer this concept was discussed heavily in the rebuttal. We only state, as the reviewer summarized above very well, that EHMT2 restricts WNT signaling and if EHMT2 is lost or inhibited then excess Wnt signaling can lead to loss of TPC function. Any speculation regarding WNT signaling thresholds occurs in the discussion and we have reviewed that section carefully to ensure that it is presented as such. Specifically, WNT signaling thresholds are only mentioned twice in the second to last paragraph of the discussion and only in a manner postulating their existence as a means to explain discrepancies between published work.

7. It is unfortunate that the reviewers were not able to confirm the RUVBL2 and B-Catenin interaction in tumorspheres or TPCs, and that mouse cell lines do not behave like TPCs in order for them to be used as a model. This raises the question then, if the A549 cells used for the mechanistic experiments contain "the relevant population to progeny relationship found in de novo kp tumors". Do A549 cells behave similarly to TPCs in organoid culture in response to G9a inhibition? This is important for the connection of the proposed mechanism to the phenotypes observed in the mouse. If the mechanistic relationship cannot be tested in the phenotypic model(s), then the phenotype should be confirmed in the mechanistic model, otherwise we cannot connect the mechanism to the phenotype even if they are 'operating in the same direction' (positively corelating). As mentioned earlier in this discussion, Zhang et al. found that Wnt signaling was inhibited by G9a inhibition and EHMT2 shRNA in cultures and xenografts of human cells including A549 cells. Whilst this study should not be considered the last word on this topic, it highlights the importance for the authors to test to see if their phenotype can be reproduced in this system used to elucidate their proposed mechanism.Whilst the authors have convincingly shown that Wnt signaling is modulated upwards in response to G9a inhibition in tumors and AT2 cells, a stronger connection to the proposed mechanisms is required for publication.

We refer the reviewer to several mechanistic assays conducted in TPCs: including proximity ligation assay between RUVBL2 and β catenin, confirming interaction and ChIP-qPCR of RUVBL2 on TCF4 motifs in AT2 genes, proving that EHMT2 directly controls *Tcf4*-containing AT2 gene expression through RUVBL2-mediated repression of β-catenin transcription. Furthermore, we also want to remind the reviewer of the observed changes in RUVBL2 levels upon EHMT2 inhibition in tumorspheres, indicating a direct relationship between EHMT2 and RUVBL2 (Figure 5c-g). Therefore, we disagree with the general statement that “the mechanistic relationship cannot be confirmed in the phenotypic model”. We used A549 as a complementary approach for experiments that are technically not feasible in TPC. Please also refer to our response for the first point raised by the reviewer (R2 Specific Point 1).

8. It is helpful that the authors provide pharmacokinetic data showing consistent plasma concentration of their inhibitor following. We take it that the G9a inhibitor is highly selective over other histone methyltransferases which has been amply demonstrated in the original literature. Whilst direct evidence of G9a inhibition in vivo would (through histone methylation) would be greatly convincing, we appreciate that confirmation of the main phenotypes of increases of wnt signaling and AT2 gene expression through genetic deletion makes it likely that G9a inhibitor is functioning as expected in vivo. In the absence of confirming whether G9a inhibition reduces the stem cell capacity by serial passaging of inhibited AT2 cell cultures, the text changes made by the authors is appropriate.

We are thankful for the input and approval of our edits.

9. The author's test if there are changes in the proportion of AT1 markers in their tumorsphere assays with G9a inhibition, as they see with their alveolar spheres. Here they do not see a significant change AT1 cell markers, highlight differences between how tumor cells and AT2 cells respond to G9a inhibition, which should be commented on in the discussion.

We appreciate the reviewer’s comment and have now commented about these differences in the text.

“In contrast to the TPC subset, we observed significant changes in the proportion of AT1 cells, consistent with the reduced plasticity observed in AT2 cells.”

The authors have already addressed wording changes to the descriptions of their alveolar sphere cultures to reflect that AT2 to AT1 is the normal differentiation process and not an example of trans-differentiation.

We are thankful for the input and approval of our edits.

Whilst the authors' results are consistent with the explanation that a change in AT2 to AT1 differentiation, the small size of the alveospheres does suggest that changes in proliferation could also contribute to these differences, and really should be explored by the authors. The immunohistochemistry would also be extremely informative as well, what to the AT1/AT2 cells look like?

We are relieved to read that the reviewer agrees that our results are consistent with the conclusion of the differentiation phenotype. With respect to the proliferation effects, we openly include data on sphere size and the effects on proliferation and apoptosis, in addition to the differentiation phenotypes (Figure 3, including the IF images, plus quantitation and Figure 6 with supplements with quantitation of the IF images). To further support our reasoning, we include the following statement with references regarding the relationship between proliferation and differentiation:

“Reduced proliferation and cell death were previously associated with cell differentiation (Domen and Weissman, 1999; Ruijtenberg and van den Heuvel, 2016).”

The authors point out that it is not possible to perform secondary transplantations with their inducible due to the re-expression of G9a in vivo. As the effects on self-renewal cannot be experimentally verified, the authors should modify their claims in the manuscript.

To clarify, the in vivo tumor formation studies using shEHMT2 are performed following secondary transplantation. To alleviate any concerns of interpretation that reviewer raises, we modified the section heading as below:

Original:

EHMT2 is required for in vivo tumor self-renewal

New:

EHMT2 is required for in vivo tumor growth

Reviewer #3 (Recommendations for the authors):1. The authors have addressed some of my previous comments. However, I find their responses to comments 12 and 13 vague and inadequate

We are disheartened to read this response from R3.

Comment 12 seemed to us to reflect thoughts of the reviewer regarding aspects of the field and did not contain a clear question. We agreed with the reviewer’s summary of the field and the speculation they proposed.

Comment 13 inquired about AT2 cells in our tumorsphere cultures, which we viewed as a misunderstanding of the technical aspects of the assay, which we clarified for the reviewer.

2. Provide one or more citations for the claim that, "…multiple lines of evidence support the expression of LGR5 in the epithelial cell compartment." or, if that is not possible, then omit reference to LGR5 expression.

The reviewer misunderstood our response to his query regarding LGR5 expression that we detect in our studies. The “multiple lines of evidence” is in reference to our own data contained in this paper. The original response is copied below for reference. Given the paucity of data on this topic and the fact that LGR5 is a well-established WNT target gene, we are compelled to retain this information.

Response from first round of reviews:

The reviewer is referring to Figure 6D, where expression of Axin2, Lgr4 and Lgr5 is used to confirm Wnt pathway activation in primary AT2-derived murine spheres following G9a inhibition. We presume that the reviewer is basing his statements of lack of Lgr5 on the data published by Lee et al. Cell 2017, where Lgr5 was not detected in the epithelial population by single cell RNA-seq. Because of dropouts, genes undetected in scRNA-seq assays cannot be considered as a confirmation of lack of expression, especially for lowly expressed genes. Indeed, expression of Lgr5 in our own bulk RNA-seq dataset was very low (2-17 reads out of 30-40M reads per sample), which likely means that Lgr5 cannot be detected in standard scRNA-seq assays. Although Lgr5 is a low-expressed gene, it is readily observable by qPCR, most likely due to the amplification steps included in this procedure. While Axin2 and Lgr4 increases suffice to demonstrate wnt pathway activation, we would like to retain Lgr5 as multiple lines of evidence support its expression in the epithelial cell compartment.